# SONAR: A SYNTHETIC AI-AUDIO DETECTION FRAMEWORK AND BENCHMARK

## ABSTRACT

Recent advances in Text-to-Speech (TTS) and Voice-Conversion (VC) using generative Artificial Intelligence (AI) technology have made it possible to generate high-quality and realistic human-like audio. This introduces significant challenges to distinguishing AI-synthesized speech from the authentic human voice and could raise potential issues of misuse for malicious purposes such as impersonation and fraud, spreading misinformation, deepfakes, and scams. However, existing detection techniques for AI-synthesized audio have not kept pace and often exhibit poor generalization across diverse datasets. In this paper, we introduce SONAR, a synthetic AI-Audio Detection Framework and Benchmark, aiming to provide a comprehensive evaluation for distinguishing cutting-edge AI-synthesized auditory content. SONAR includes a novel evaluation dataset sourced from 9 diverse audio synthesis platforms, including leading TTS providers and state-of-the-art TTS models. It is the first framework to uniformly benchmark AI-audio detection across both traditional and foundation model-based deepfake detection systems. Through extensive experiments, (1) we reveal the generalization limitations of existing detection methods and demonstrate that foundation models exhibit stronger generalization capabilities, which can be attributed to their model size and the scale and quality of pretraining data. (2) Our evaluation of the generalization across languages suggests that speech foundation models demonstrate robust cross-lingual generalization capabilities, maintaining strong performance across diverse languages despite being fine-tuned solely on English speech data. This finding also suggests that the primary challenges in audio deepfake detection are more closely tied to the realism and quality of synthetic audio rather than language-specific characteristics. (3) We also explore the effectiveness and efficiency of few-shot fine-tuning in improving generalization, highlighting its potential for tailored applications, such as personalized detection systems for specific entities or individuals. *Code and dataset are available at* `https://anonymous.4open.science/r/SONAR`

## 1 INTRODUCTION

Recent advances in Text-to-Speech (TTS) and Voice-Conversion (VC) using Artificial Intelligence (AI) technology have made it possible to generate high-quality and realistic human-like audio efficiently (Vyas et al., 2023; Ye et al., 2024; Casanova et al., 2022; Wang et al., 2023). This introduces significant challenges in distinguishing AI-synthesized speech from the authentic human voice and could raise potential misuse for malicious purposes such as impersonation and fraud, spreading misinformation, and scams. For example, a deep fake AI voice of the US President Joe Biden was recently utilized in robocalls to advise them against voting[1], demonstrating how deepfakes can significantly manipulate public opinions and influence presidential elections. In response to such risks, the US Federal Communications Commission (FCC) now deems robot calls for election as illegal, which underscores the urgent need for enhanced detection of AI-synthesized audio.

While TTS models are advancing rapidly, AI-synthesized audio detection techniques are not keeping pace. First, previous studies (Müller et al., 2022; Zang et al., 2024) have highlighted the lack of

---

[1]`https://www.cnn.com/2024/01/22/politics/fake-joe-biden-robocall/index.html`

generalization and robustness in these detection methods. Second, existing detection models (Jung et al., 2022; Zang et al., 2024; Tak et al., 2021b;a; Lavrentyeva et al., 2019) often take advantage of different audio features and evaluation datasets, complicating the comparison of their detection effectiveness. Third, a comprehensive evaluation to determine the effectiveness of these detection methods against the latest TTS models has not been conducted. This gap in research leaves a significant challenge in developing reliable detection techniques that can effectively counter the growing sophistication of AI-generated audio.

To address the aforementioned research gap and explore the strengths and limitations of existing AI-synthesized audio detection methods, especially those with increasingly advanced TTS models, we present a synthetic AI-Audio Detection Framework and Benchmark, coined as SONAR. This framework aims to provide a comprehensive evaluation for distinguishing state-of-the-art AI-synthesized auditory content. Our study benchmarks the state-of-the-art fake audio detection models using a newly collected fake audio dataset that includes a variety of synthetic speech audios sourced from diverse cutting-edge TTS providers and TTS models. We further investigate the potential of enhancing the generalization capabilities of these detection models from different perspectives. The main contributions of our work can be summarized as follows.

- We introduce a novel evaluation dataset specifically designed for audio deepfake detection. This dataset is sourced from 9 diverse audio synthesis platforms, including those from leading TTS service providers and state-of-the-art TTS models. To the best of our knowledge, this dataset is by far the largest collection of fake audio generated by the latest TTS models.

- SONAR is the first comprehensive framework to benchmark AI-audio detection uniformly across advanced TTS models. It covers 5 state-of-the-art traditional and 6 foundation-model-based audio deepfake detection models.

- Leveraging SONAR, we conduct extensive experiments to analyze the generalizability limitations of current detection methods. Our findings reveal that foundation models demonstrate stronger generalization capabilities than traditional models. We further explore factors that may contribute to this improved generalization, such as model size and the scale and quality of pre-training data.

- Our evaluation of the generalization across languages suggests that speech foundation models demonstrate robust cross-lingual generalization capabilities, maintaining strong performance across diverse languages despite being fine-tuned solely on English speech data. This finding also suggests that the primary challenges in audio deepfake detection are more closely tied to the realism and quality of synthetic audio rather than language-specific characteristics.

- We further explore the potential of few-shot fine-tuning to enhance the generalization of detection models. Our empirical results demonstrate the effectiveness and efficiency of this approach, highlighting its potential for tailored applications, such as personalized detection systems for specific entities or individuals.

## 2 EVALUATION DATASET GENERALIZATION AND COLLECTION

Leveraging a set of diverse and high-quality speech data synthesis APIs and models, we create an evaluation dataset for synthetic AI-audio detection. Our approach incorporates two strategies: data generation and data collection. Our dataset includes AI-generated speech and audio from nine distinct sources. We perform speech data generation using one cutting-edge TTS service provider, OpenAI, and two open-sourced APIs, xTTS (Casanova et al., 2024) and AudioGen (Kreuk et al., 2022). For speech data collection, we leverage six state-of-the-art TTS models including Seed-TTS (Anastassiou et al., 2024), VALL-E (Wang et al., 2023), PromptTTS2 (Leng et al., 2023), NaturalSpeech3 (Ju et al., 2024), VoiceBox (Le et al., 2024), FlashSpeech (Ye et al., 2024). Table 1 presents the details of our dataset generated by different audio generation models. We next detail our methods of generating and collecting these datasets.

**Data generation**. Our dataset generation involves OpenAI, xTTS, and AudioGen. Specifically, OpenAI currently provides voice choices from 6 different speakers. Using ChatGPT,we generate 100 different text prompts of varying lengths for each speaker, resulting in a total of 600 synthetic speech audios. xTTS supports synthetic speech generation given text prompts and reference speech.

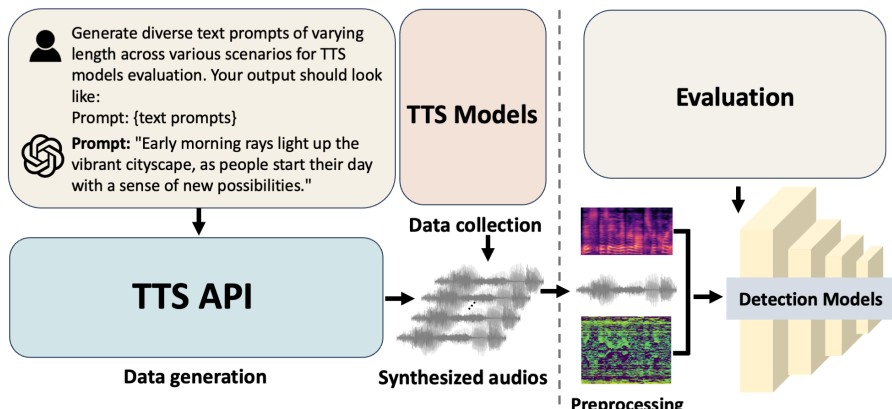

Figure 1: Overview of SONAR. **Left**: Audio deepfake data generation and collection. **Right**: Benchmark evaluation.

We select 6 speakers from the LibriTTS dataset (Zen et al., 2019) as the reference speech and also generate 600 text prompts with ChatGPT for each speaker, resulting in 600 synthetic speech audios. To evaluate whether speech detection models can generalize to AI-synthesized environmental sound, we also include a subset consisting of AI-synthesized environmental sound. AudioGen can generate the corresponding environmental sound given a textual description of the acoustic scene. With AudioGen, we use ChatGPT to generate 100 text descriptions of the environment and background and obtain 100 AI-synthesized environmental sounds. Figure 1 (left) illustrates the data generation and collection process.

**Data collection**. To evaluate the effectiveness of various detection systems against the state-of-the-art TTS models, we also collect fake speech audio from Seed-TTS, VALL-E, PromptTTS2, NaturalSpeech3, VoiceBox, and FlashSpeech. Seed-TTS provides a test dataset[2] consisting of fake audio samples generated by it. Due to the unavailability of pre-trained weights of the other 5 models, we extract the synthesized speech data directly from their demo pages. Specifically, speech audios from VALL-E include variations in emotions and acoustic environment. PromptTTS2 presents fake audio samples with various attributes such as gender, speed, pitch, volume, and timbre. NaturalSpeech3 also includes fake audio samples generated with various attributes such as speeds and emotions and contains fake speech audio samples obtained with voice conversion. VoiceBox provides fake audio samples that feature cross-lingual and expressive audio styles. FlashSpeech includes a set of high-quality fake audios obtained both from speech generation and voice conversion.

To summarize, leveraging the outlined details, we generate and collect a comprehensive evaluation dataset, encompassing a total of 2274 AI-synthesized audio samples produced by various TTS models. To the best of our knowledge, our dataset is by far the largest collection of fake audio generated by the latest TTS models. Note our motivation for collecting this dataset is for evaluation purposes. Additionally, we only include fake audio samples in this dataset since genuine audio samples can be

---

[2] https://github.com/BytedanceSpeech/seed-tts-eval

Table 1: Overview of our dataset with fake audios generated by various models. AudioGen lacks speaker and language information as it focuses on environmental sounds. The trainset sizes for OpenAI and Seed-TTS are unavailable due to the use of proprietary data. * denotes the samples that are directly collected from their demo page or provided test set due to the unavailability of their model checkpoints.

| Model | Samples | Avg duration (s) | Avg. pitch (Hz) | Std. pitch (Hz) | Languages | Trainset size(H) | Year |
|---|---|---|---|---|---|---|---|
| PromptTTS2* | 25 | 9.86 | 126.49 | 46.27 | English | 44K | 2023 |
| NaturalSpeech3* | 32 | 5.25 | 143.86 | 53.94 | English | 60K | 2024 |
| VALL-E* | 95 | 4.86 | 133.41 | 56.54 | English | 60K | 2023 |
| VoiceBox* | 104 | 10.28 | 114.09 | 37.89 | English, German, French, Portuguese, Polish, Spanish | 60K | 2023 |
| FlashSpeech* | 118 | 7.57 | 129.30 | 54.77 | English | 44.5K | 2024 |
| AudioGen | 100 | 5.00 | 199.45 | 72.94 | - | 7K | 2022 |
| xTTS | 600 | 5.67 | 164.67 | 95.20 | English | 2.7K | 2023 |
| Seed-TTS* | 600 | 4.91 | 117.28 | 36.85 | English, Mandarin | - | 2024 |
| OpenAI | 600 | 4.11 | 126.89 | 54.89 | English | - | 2024 |

easily collected from various sources (e.g., internet, self-recording, publicly available datasets, etc.). However, for convenience of evaluation, we also provide an equal number of real speech audio data sampled from the LibriTTS (Zen et al., 2019) clean-test set. We believe the collected dataset can serve as a valuable asset for evaluating existing audio deepfake detection models.

# 3 BENCHMARKING AI-AUDIO DETECTION MODELS

In this section, we first detail the model, dataset, and evaluation metrics setup for benchmarking. Then, we present the results of evaluating detection models on existing audio deepfake datasets to assess their generalizability across datasets. We next benchmark their detection performance on our proposed dataset and provide analysis for potential model generalization improvement.

## 3.1 BENCHMARKING SETUP

**Model architectures**. SONAR incorporates 11 models, including 5 state-of-the-art traditional audio deepfake detection models featuring various levels of input feature abstraction and 6 foundation models. Specifically, for the former, SONAR includes (1) AASIST (Jung et al., 2022), which processes raw waveform directly and utilizes graph neural networks and incorporates spectro-temporal attention mechanisms. (2) RawGAT-ST (Tak et al., 2021a), which employs spectral and temporal sub-graphs along with a graph pooling strategy. (3) RawNet2 (Tak et al., 2021b), which is a hybrid model combining CNN and GRU.(4) Spectrogram(Spec.)+ResNet (Zang et al., 2024), which transforms the audio to linear spectrogram using a 512-point Fast Fourier Transform (FFT) with a hop size of 10 ms. The spectrogram is then inputted into ResNet18 (He et al., 2016). (5) LFCC-LCNN (Lavrentyeva et al., 2019), which converts audio into Linear-Frequency Cepstral Coefficients (LFCC) for input into a CNN model. Specifically, 60-dimensional LFCCs are extracted from each utterance frame, with frame length set to 20ms and hop size 10ms. It extracts speech embedding directly from raw audio. These models collectively cover a broad spectrum of feature types and architectures, facilitating a detailed examination of their performance in deepfake audio detection applications. For *foundation models*, SONAR includes (1) Wave2Vec2 (Baevski et al., 2020), which is pre-trained on 53k hours of unlabeled speech data. (2) Wave2Vec2BERT (Barrault et al., 2023), which is pre-trained on 4.5M hours of unlabeled speech data covering more than 143 languages. (3) HuBERT (Hsu et al., 2021), which is pretrained-on 60k hours of speech data. (4) CLAP (Elizalde et al., 2023), who is trained on a variety of audio-text pairs. (5) Whisper-small (Radford et al., 2023), and (6) Whisper-large (Radford et al., 2023). Both Whispers are pre-trained on 680K hours of speech data covering 96 languages.

**Public datasets for training and testing**. We consider five benchmark datasets for deepfake audio detection model training and testing as they are commonly used in the literature (Kawa et al., 2022b;a). **Wavefake** (Frank & Schönherr, 2021) collects deepfake audios from six vocoder architectures, including MelGAN (Kumar et al., 2019), FullBand-MelGAN, MultiBand-MelGAN (Yang et al., 2021), HiFi-GAN (Kong et al., 2020a), Parallel WaveGAN (Yamamoto et al., 2020), and WaveGlow (Prenger et al., 2019). It consists of approximately 196 hours of generated audio files derived from the LJSPEECH (Ito & Johnson, 2017) dataset. Similar to wavefake, **LibriSe-Voc** (Sun et al., 2023) collects deepfake audios from six state-of-the-art neural vocoders including WaveNet(Van Den Oord et al., 2016), WaveRNN (Kalchbrenner et al., 2018), Mel-GAN (Yang et al., 2021), Parallel WaveGAN (Yamamoto et al., 2020), WaveGrad (Chen et al., 2020a) and DiffWave (Kong et al., 2020b) to generate speech samples derived from the widely used LibriTTS speech corpus (Zen et al., 2019), which is often utilized in text-to-speech research. Specifically, it consists of a total of 208.74 hours of synthesized samples. **In-the-wild** (Müller et al., 2022) comprises genuine and deepfake audio recordings of 58 politicians and other public figures gathered from publicly accessible sources, including social networks and video streaming platforms. MLAAD (Müller et al., 2024) consists of fake audios created using 82 TTS models, covering 38 different languages. We use this dataset to evaluate the cross-lingual generalization of different detection models. To further evaluate generalization capabilities on voice-converted audio, we utilize a subset of the **ASVSpoof2019** development set (Wang et al., 2020), which contains synthetic speech generated through voice conversion techniques. All input audios are resampled to a 16kHz sampling rate and converted into raw waveforms consisting of 64,000 samples (approximately 4

Table 2: Generalization across existing audio deepfake datasets. All models are trained/finetuned on the Wavefake training set. Green and orange indicate the best and second-best performance, respectively.

| Model | Wavefake | | | LibriSeVoc | | | In-the-wild | | |
|---|---|---|---|---|---|---|---|---|---|
| | Accuracy | AUROC | EER(%) | Accuracy | AUROC | EER(%) | Accuracy | AUROC | EER(%) |
| LFCC-LCNN | 0.9984 | 0.9999 | 0.153 | 0.7429 | 0.8239 | 25.71 | 0.5 | 0.4786 | 99.2 |
| Spec.+ResNet | 0.9924 | 0.9924 | 0.076 | 0.7577 | 0.8495 | 24.233 | 0.4685 | 0.4723 | 53.148 |
| RawNet2 | 0.9416 | 0.9592 | 5.839 | 0.5119 | 0.5332 | 48.807 | 0.5321 | 0.5393 | 46.792 |
| RawGATST | 0.9988 | 0.9999 | 0.115 | 0.8307 | 0.9203 | 16.925 | 0.6418 | 0.7015 | 35.816 |
| AASIST | 0.9992 | 0.9999 | 0.076 | 0.886 | 0.9534 | 11.397 | 0.7272 | 0.7975 | 27.277 |
| CLAP | 0.9996 | 0.9999 | 0.038 | 0.8296 | 0.9019 | 24.763 | 0.3013 | 0.2252 | 69.871 |
| Whisper-small | 0.9935 | 0.9997 | 0.649 | 0.9345 | 0.9837 | 6.551 | 0.821 | 0.9025 | 17.899 |
| Whisper-large | 0.9962 | 0.9992 | 0.381 | 0.9572 | 0.9901 | 4.279 | 0.8848 | 0.9552 | 11.518 |
| Wave2Vec2 | 0.9874 | 0.9987 | 1.259 | 0.9705 | 0.9953 | 2.953 | 0.8733 | 0.9323 | 12.669 |
| HuBERT | 0.9931 | 0.9996 | 0.687 | 0.986 | 0.9991 | 1.401 | 0.9164 | 0.9653 | 8.362 |
| Wave2Vec2BERT | 0.9996 | 0.9999 | 0.038 | 0.9902 | 0.9991 | 0.984 | 0.9232 | 0.979 | 7.676 |

seconds). Audios longer than 4 seconds are randomly trimmed, while those shorter than 4 seconds are repeated and padded to meet the 4-second duration.

For LibriSeVoc, we follow the official train-validation-test splits, which are approximately 60%, 20%, and 20%, respectively. For Wavefake, we partition the data generated by each vocoder into training, validation, and testing subsets at ratios of 70%, 10%, and 20%, respectively. To address the class imbalance and mitigate potential evaluation bias, we further downsample LibriSeVoc and WaveFake test datasets, and In-the-Wild datasets, resulting in a balanced dataset with a real-to-fake ratio of 1:1.

**Evaluation metrics**. To provide a comprehensive evaluation of the detection performance of audio deepfake models, we adopt (1) *Equal Error Rate* (EER), which is defined as the point on the ROC curve, where the false positive rate (FPR) and false negative rate (FNR) are equal and is commonly used to assess the performance of binary classifications tasks, with lower values indicating better detection performance. (2) *Accuracy* evaluates the overall correctness of the detection model's predictions and is defined as the ratio of correctly predicted data to the total data. To ensure consistency with the EER and provide more intuitive results, we set the threshold for accuracy at the EER point, meaning the accuracy reflects the model's performance when the FPR equals the FNR. (3) *AUROC* (Area Under the Receiver Operating Characteristic) provides a measure of the model's ability to distinguish between classes across different decision thresholds, providing a more comprehensive view of its discriminative power across varying conditions. An AUROC score of 1.0 indicates perfect classification, while a score of 0.5 indicates performance no better than random guessing.

Note that the test datasets are class-balanced, and the accuracy score is calculated using the EER threshold. Thus, we omit F1, precision, and recall scores from our evaluation results in the paper, though SONAR provides these metrics as well.

## 3.2 RESULTS AND ANALYSIS

### 3.2.1 HOW WELL CAN DETECTION MODELS GENERALIZE ACROSS DATASETS?

We first train all models on Wavefake training dataset and then evaluate the models on its own test set, LibriSeVoc test set, and In-the-wild dataset. Table 2 presents the evaluation results. Particularly, we make the following interesting observations.

**Speech foundation models exhibit stronger generalizability**. As shown in Table 2, when evaluated on the test set of Wavefake, all models demonstrate near-perfect performance across the three metrics. This can be attributed to the similarity between the test set and the training data. However, when tested on the LibriSeVoc and In-the-wild datasets, models such as LFCC-LCNN, Spec.+ResNet, RawNet2, RawGATST, and AASIST struggle to generalize effectively. This performance gap indicates significant overfitting to the training data, despite these models being specifically designed for audio deepfake detection tasks. In contrast, speech foundation models consistently display stronger generalizability. Notably, Wave2Vec2BERT achieves the highest generalizability, which may be attributed to its large-scale and diverse pretraining data. Pretrained on 4.5 million hours of unlabeled audio in more than 143 languages, Wave2Vec2BERT benefits from both scale and diversity. This suggests that a well-designed self-supervised model trained on diverse speech data can extract general and discriminative features, making it more applicable across different datasets for audio deepfake detection. It is important to note that CLAP, unlike other speech

Table 3: Evaluation on SONAR dataset. Green and orange indicate the best and second-best performance, respectively.

(a) Accuracy (↑).

| Model | PromptTTS2 | NaturalSpeech3 | VALL-E | VoiceBox | FlashSpeech | AudioGen | xTTS | Seed-TTS | OpenAI | Average |
|---|---|---|---|---|---|---|---|---|---|---|
| LFCC-LCNN | 0.5200 | 0.7500 | 0.6211 | 0.8462 | 0.7034 | 0.4600 | 0.7433 | 0.3058 | 0.5000 | 0.6055 |
| Spec.+ResNet | 0.5600 | 0.5000 | 0.5684 | 0.5481 | 0.6356 | 0.6800 | 0.8450 | 0.4167 | 0.6783 | 0.6036 |
| RawNet2 | 0.6800 | 0.3125 | 0.4211 | 0.5385 | 0.4915 | 0.2600 | 0.6533 | 0.3733 | 0.3500 | 0.4534 |
| RawGATST | 0.8000 | 0.5312 | 0.6842 | 0.8173 | 0.5424 | 0.2400 | 0.6567 | 0.5833 | 0.4900 | 0.5939 |
| AASIST | 0.8400 | 0.5312 | 0.7789 | 0.8750 | 0.6610 | 0.6900 | 0.7300 | 0.6567 | 0.5150 | 0.6975 |
| CLAP | 0.5600 | 0.4688 | 0.6421 | 0.5288 | 0.6017 | 0.2500 | 0.4800 | 0.4000 | 0.3233 | 0.4727 |
| Whisper-small | 0.8800 | 0.5625 | 0.7158 | 0.7404 | 0.5678 | 0.8000 | 0.8050 | 0.5983 | 0.1883 | 0.6509 |
| Whisper-large | 1.000 | 0.6562 | 0.7895 | 0.7885 | 0.7288 | 0.8400 | 0.9033 | 0.5933 | 0.2900 | 0.7322 |
| Wave2Vec2 | 0.9600 | 0.6875 | 0.8210 | 0.9327 | 0.8136 | 0.9900 | 0.7333 | 0.8683 | 0.5175 | 0.8138 |
| HuBERT | 1.0000 | 0.7500 | 0.9158 | 0.9712 | 0.9407 | 1.0000 | 0.8767 | 0.8900 | 0.5658 | 0.8789 |
| Wave2Vec2BERT | 1.0000 | 0.9062 | 0.9474 | 0.9712 | 0.9237 | 0.9700 | 0.9867 | 0.6017 | 0.7833 | 0.8989 |

(b) AUROC (↑).

| Model | PromptTTS2 | NaturalSpeech3 | VALL-E | VoiceBox | FlashSpeech | AudioGen | xTTS | Seed-TTS | OpenAI | Average |
|---|---|---|---|---|---|---|---|---|---|---|
| LFCC-LCNN | 0.5696 | 0.7666 | 0.6967 | 0.9106 | 0.7945 | 0.4559 | 0.8163 | 0.2452 | 0.0967 | 0.5947 |
| Spec.+ResNet | 0.6064 | 0.4941 | 0.6217 | 0.5858 | 0.6891 | 0.7293 | 0.9205 | 0.4003 | 0.7450 | 0.6436 |
| RawNet2 | 0.6944 | 0.2422 | 0.3695 | 0.6210 | 0.5203 | 0.3030 | 0.7210 | 0.3120 | 0.2940 | 0.4530 |
| RawGATST | 0.8704 | 0.5439 | 0.7490 | 0.8989 | 0.5742 | 0.2050 | 0.7317 | 0.6065 | 0.4795 | 0.6288 |
| AASIST | 0.9248 | 0.6172 | 0.8479 | 0.9433 | 0.7485 | 0.7466 | 0.8265 | 0.6893 | 0.5259 | 0.7633 |
| CLAP | 0.5712 | 0.4434 | 0.7223 | 0.5155 | 0.6533 | 0.1777 | 0.5114 | 0.3544 | 0.2407 | 0.4655 |
| Whisper-small | 0.9776 | 0.5762 | 0.8050 | 0.8400 | 0.6446 | 0.8284 | 0.8915 | 0.6326 | 0.108 | 0.7004 |
| Whisper-large | 1.0000 | 0.6992 | 0.9063 | 0.8552 | 0.7933 | 0.8926 | 0.9690 | 0.6558 | 0.2327 | 0.7782 |
| Wave2Vec2 | 0.9952 | 0.7515 | 0.8751 | 0.9674 | 0.8438 | 0.9987 | 0.7931 | 0.9205 | 0.4881 | 0.8482 |
| HuBERT | 1.0000 | 0.8174 | 0.9719 | 0.9953 | 0.9871 | 1.0000 | 0.9496 | 0.9531 | 0.5585 | 0.9148 |
| Wave2Vec2BERT | 1.0000 | 0.9658 | 0.9860 | 0.9906 | 0.9666 | 0.9826 | 0.9980 | 0.6165 | 0.8607 | 0.9290 |

(c) EER(%) (↓).

| Model | PromptTTS2 | NaturalSpeech3 | VALL-E | VoiceBox | FlashSpeech | AudioGen | xTTS | Seed-TTS | OpenAI | Average |
|---|---|---|---|---|---|---|---|---|---|---|
| LFCC-LCNN | 48.000 | 25.000 | 37.895 | 15.385 | 29.661 | 54.000 | 25.667 | 69.5 | 99.333 | 44.938 |
| Spec.+ResNet | 44.000 | 50.000 | 43.158 | 45.192 | 36.441 | 32.000 | 15.500 | 58.333 | 32.167 | 39.643 |
| RawNet2 | 32.000 | 68.750 | 57.895 | 46.154 | 50.848 | 74.000 | 34.667 | 62.667 | 65.000 | 54.665 |
| RawGATST | 20.000 | 46.875 | 31.580 | 18.269 | 45.763 | 76.000 | 34.330 | 41.667 | 51.000 | 40.609 |
| AASIST | 16.000 | 46.875 | 22.105 | 12.500 | 33.898 | 31.000 | 27.000 | 34.333 | 48.500 | 30.246 |
| CLAP | 44.000 | 53.125 | 35.789 | 47.115 | 39.831 | 75.000 | 52.000 | 60.000 | 67.667 | 52.725 |
| Whisper-small | 12.000 | 43.750 | 28.421 | 25.962 | 43.220 | 20.000 | 19.500 | 40.167 | 81.167 | 34.910 |
| Whisper-large | 0.000 | 34.375 | 21.053 | 21.154 | 27.119 | 16.000 | 9.667 | 40.667 | 71.000 | 26.782 |
| Wave2Vec2 | 4.000 | 31.250 | 17.895 | 6.731 | 18.644 | 1.000 | 26.667 | 13.167 | 48.333 | 18.632 |
| HuBERT | 0.000 | 25.000 | 8.421 | 2.885 | 5.932 | 0.000 | 12.333 | 11.000 | 43.500 | 12.119 |
| Wave2Vec2BERT | 0.000 | 9.375 | 5.263 | 2.885 | 7.627 | 3.000 | 1.333 | 39.833 | 21.667 | 10.109 |

foundation models, does not generalize well across datasets. This is likely due to its primary focus on environmental audio data during pretraining, resulting in the extraction of irrelevant features for speech audio. This observation underscores that not all foundation models are equally suited for audio deepfake detection tasks.

**Generalizability may increase with model size.** In Table 2, it can be observed that Whisper-large always outperforms Whisper-small across all three datasets. In particular, on the LibriSeVoc test set, Whisper-large achieves accuracy, AUROC, and EER of 0.9572, 0.9901, 4.279%, respectively, which improves by 2.27%, 0.64%, and 2.272%, than that of Whisper-small. This trend is more evident in the In-the-wild dataset, which is closer to real-world scenarios since this dataset consists of speech data sourced from the internet. Specifically, Whisper-large achieves accuracy, AUROC, and EER of 0.8848, 0.9552, and 11.518%, respectively, which improves by 6.381%, 5.27%, and 6.381%, than that of Whisper-small. Further investigation will be made in Section 3.2.3

### 3.2.2 RESULTS ON SONAR DATASET

We further evaluate all detection models on the proposed dataset. Table 3a, Table 3b, and Table 3c present the accuracy, AUROC, and EER of different detection models on our proposed SONAR dataset as described in Sec 2.

**Speech foundation models can better generalize on the SONAR dataset, but still not good enough.** As presented in Table 3a, speech foundation models again exhibit better generalizability on the fake audio samples generated by the latest TTS models. For instance, AASIST achieves 0.6975 average accuracy across audios generated by cutting-edge TTS models, which is the best performance among the traditional detection models. In contrast, speech foundation models Whisper-large, Wave2Vec2, HuBERT, and Wave2Vec2BERT achieve an average accuracy of 0.7322, 0.788, 0.8789, and 0.8989, respectively, which is higher than AASIST by 3.47%, 9.05%, 18.14%, and

20.14%, respectively. More specifically, even though Wave2Vec2BERT and HuBERT are only fine-tuned on Wavefake dataset, for PromptTTS2, VALL-E, VoiceBox, FalshSpeech, AudioGen, and xTTS, Wave2Vec2BERT can reach accuracies of 1.0, 0.9062, 0.9474, 0.9712, 0.9237, 0.97, and 0.9867, respectively, and HuBERT can achieve 1.0, 0.9158, 0.9712, 0.9407, 1.0 0.8767, and 0.89, respectively, demonstrating their potential capability of extract more distinguishable features compared to other models. It is also worth noting that Wave2Vec2BERT achieves an accuracy of 0.9062 on NaturalSpeech3, while all other models can only reach that $\leq 0.75$.

**It is still challenging for detection models to correctly classify synthesized audio samples, especially those generated by the most advanced TTS service providers.** While Wave2Vec2BERT achieves an overall average accuracy of 0.8989, it only reaches 0.6017 on Seed-TTS and 0.7833 on OpenAI. A similar pattern is also evident with HuBERT, Wave2Vec2, Whisper-large, and Whisper-small, which achieve just 0.5658, 0.4342, 0.29, and 0.1883 accuracy on OpenAI, respectively. This performance disparity is likely due to OpenAI and Seed-TTS having more advanced model architectures and being trained on proprietary, self-collected data, leading to higher-quality and more realistic speech generation. We will explore potential strategies to enhance their detection performance in Section 3.2.4. Overall, these results not only indicate that no single model consistently outperforms across all datasets but also underscore the ongoing difficulty in detecting synthesized audio from cutting-edge TTS systems, especially those developed by the most advanced TTS service providers. This highlights a huge gap between the rapid evolution of TTS technologies and the effectiveness of current audio deepfake detection methods, emphasizing the urgent need for the development of more robust and reliable detection algorithms.

Additionally, it is noteworthy that, compared to speech foundation models, the accuracy of all five traditional detection models on the AudioGen dataset, which consists of synthesized environmental sounds, remains relatively low. Specifically, LFCC-LCNN, Spec.+ResNet, RawNet2, RawGATST, and AASIST achieve accuracies of 0.46, 0.68, 0.26, 0.24, and 0.69, respectively. In contrast, Whisper-small, Whisper-large, Wave2Vec2, HuBERT, and Wave2VecBERT attain significantly higher accuracies of 0.8, 0.84, 0.99, 1.0, and 0.97, respectively. This discrepancy may be due to traditional detection models being trained exclusively on speech data, which limits their generalization to audio from different distributions. In comparison, foundation models demonstrate greater robustness to out-of-distribution audio samples. An exception to this is CLAP, which is an audio foundation model pre-trained on a variety of environmental audio-text pairs and only achieves an accuracy of 0.25 on AudioGen. Similar to previous results, it's possibly due to the fact that its full-weight fine-tuning on speech data may have compromised its ability to effectively recognize environmental sounds, resulting in poor performance.

### 3.2.3 CAN GENERALIZABILITY INCREASE WITH MODEL SIZE?

Building on the observation that Whisper-large consistently outperforms Whisper-small, we extend our analysis with controlled experiments on the entire Whisper model family. Specifically, the Whisper family comprises five different model sizes: Whisper-tiny, Whisper-base, Whisper-small, Whisper-medium, and Whisper-large. Table 4 presents the number of model parameters of them. Specifically, each model is fine-tuned on the Wavefake training dataset using the same hyperparameters. Our results show that as model size increases, the generalizability of the models improves as well.

Table 4: Whisper model sizes.

| Model | #Params |
|---|---|
| Whisper-tiny | 39M |
| Whisper-base | 74M |
| Whisper-small | 244M |
| Whisper-medium | 769M |
| Whisper-large | 1550M |

Table 5 presents the detection performance of the Whisper models across the Wavefake, LibriSeVoc, and In-the-wild datasets. First, Whisper-tiny, despite its smaller size, still outperforms or achieves comparable detection performance to traditional detection models (recall Table 2) on the LibriSeVoc test set. This again validates the finding that foundation models exhibit stronger generalizability for audio deepfake detection tasks, even in their smallest configurations.

Second, as the model size increases from Whisper-tiny to Whisper-large, both accuracy and AUROC improve significantly across the LibriSeVoc and In-the-wild datasets. Whisper-large achieves an accuracy of 95.72% and an AUROC of 0.9901 on LibriSeVoc, surpassing Whisper-tiny by 10.07% in accuracy. A more evident pattern can be observed on the In-the-wild dataset, where Whisper-large outperforms Whisper-tiny by 38.48% in accuracy. Furthermore, the Equal Error Rate (EER)

Table 5: Generalization across existing audio deepfake datasets. All Whisper models are trained/finetuned on the Wavefake training set. Green and orange indicate the best and second-best performance, respectively.

| Model | Wavefake | | | LibriSeVoc | | | In-the-wild | | |
|---|---|---|---|---|---|---|---|---|---|
| | Accuracy | AUROC | EER(%) | Accuracy | AUROC | EER(%) | Accuracy | AUROC | EER(%) |
| Whisper-tiny | 0.9839 | 0.9985 | 1.603 | 0.8557 | 0.9307 | 14.426 | 0.498 | 0.5 | 50.203 |
| Whisper-base | 0.9908 | 0.9996 | 0.916 | 0.9163 | 0.9734 | 8.368 | 0.7398 | 0.8124 | 26.024 |
| Whisper-small | 0.9935 | 0.9997 | 0.649 | 0.9345 | 0.9837 | 6.551 | 0.821 | 0.9025 | 17.899 |
| Whisper-medium | 0.9962 | 0.9999 | 0.381 | 0.944 | 0.985 | 5.604 | 0.8572 | 0.9288 | 14.277 |
| Whisper-large | 0.9962 | 0.9992 | 0.381 | 0.9572 | 0.9901 | 4.279 | 0.8848 | 0.9552 | 11.518 |

Table 6: Evaluation on SONAR dataset. Green and orange indicate the best and second-best performance, respectively.

(a) Accuracy (↑).

| Model | PromptTTS2 | NaturalSpeech3 | VALL-E | VoiceBox | FlashSpeech | AudioGen | xTTS | Seed-TTS | OpenAI | Average |
|---|---|---|---|---|---|---|---|---|---|---|
| Whisper-tiny | 0.8000 | 0.3438 | 0.6947 | 0.6442 | 0.4661 | 0.73 | 0.6517 | 0.5067 | 0.0833 | 0.5467 |
| Whisper-base | 0.8400 | 0.4375 | 0.6947 | 0.6731 | 0.6017 | 0.6800 | 0.6550 | 0.4800 | 0.1117 | 0.5749 |
| Whisper-small | 0.8800 | 0.5625 | 0.7158 | 0.7404 | 0.5678 | 0.8000 | 0.8050 | 0.5983 | 0.1883 | 0.6509 |
| Whisper-medium | 0.96 | 0.6250 | 0.7895 | 0.8077 | 0.7119 | 0.8000 | 0.8400 | 0.5517 | 0.2183 | 0.7005 |
| Whisper-large | 1.000 | 0.6562 | 0.7895 | 0.7885 | 0.7288 | 0.8400 | 0.9033 | 0.5933 | 0.2900 | 0.7322 |

(b) AUROC (↑).

| Model | PromptTTS2 | NaturalSpeech3 | VALL-E | VoiceBox | FlashSpeech | AudioGen | xTTS | Seed-TTS | OpenAI | Average |
|---|---|---|---|---|---|---|---|---|---|---|
| Whisper-tiny | 0.9136 | 0.2998 | 0.7436 | 0.7144 | 0.4886 | 0.7660 | 0.7239 | 0.5033 | 0.0454 | 0.5776 |
| Whisper-base | 0.9296 | 0.4326 | 0.7512 | 0.7482 | 0.6548 | 0.7505 | 0.7167 | 0.5152 | 0.041 | 0.6155 |
| Whisper-small | 0.9776 | 0.5762 | 0.8050 | 0.8400 | 0.6446 | 0.8284 | 0.8915 | 0.6326 | 0.108 | 0.7004 |
| Whisper-medium | 0.9984 | 0.6279 | 0.886 | 0.8578 | 0.7950 | 0.8640 | 0.9215 | 0.5858 | 0.1567 | 0.7437 |
| Whisper-large | 1.0000 | 0.6992 | 0.9063 | 0.8552 | 0.7933 | 0.8926 | 0.969 | 0.6558 | 0.2327 | 0.7782 |

(c) EER(%) (↓).

| Model | PromptTTS2 | NaturalSpeech3 | VALL-E | VoiceBox | FlashSpeech | AudioGen | xTTS | Seed-TTS | OpenAI | Average |
|---|---|---|---|---|---|---|---|---|---|---|
| Whisper-tiny | 20.000 | 65.625 | 30.526 | 35.577 | 53.390 | 27.000 | 34.833 | 49.333 | 91.667 | 45.328 |
| Whisper-base | 16.000 | 56.250 | 30.526 | 32.692 | 36.831 | 32.000 | 34.500 | 52.000 | 88.833 | 42.811 |
| Whisper-small | 12.000 | 43.750 | 28.421 | 25.962 | 43.220 | 20.000 | 19.500 | 40.667 | 81.167 | 34.965 |
| Whisper-medium | 4.000 | 37.500 | 21.053 | 19.231 | 28.814 | 20.000 | 16.000 | 44.833 | 78.167 | 29.955 |
| Whisper-large | 0.000 | 34.375 | 21.053 | 21.154 | 27.119 | 16.000 | 9.667 | 40.167 | 71.000 | 26.726 |

decreases as the model size increases, indicating that larger models are not only more accurate but also better at minimizing both false positives and false negatives.

We also evaluate the Whisper family on the proposed SONAR dataset. Table 6a, Table 6b, Table 6c present the corresponding Accuracy, AUROC, and EER(%). A similar trend can also be observed. In Table 6a, the accuracy of the Whisper models shows a clear upward trend as the model size increases from Whisper-tiny to Whisper-large. Whisper-tiny achieves an average accuracy of 0.5467, while Whisper-large reaches the highest average accuracy of 0.7322. Notably, Whisper-large performs best on almost all datasets, particularly with TTS models such as PromptTTS2, NaturalSpeech3, VALL-E, and OpenAI, highlighting its better generalizability. Additionally, Whisper-large's performance is higher on challenging datasets like Seed-TTS and OpenAI, which are known for their high-quality synthesis. The smaller models (e.g., Whisper-tiny and Whisper-base), on the other hand, struggle to generalize effectively, particularly on datasets such as OpenAI, where the accuracy drops to 0.0833 for Whisper-tiny.

The results highlight the scalability of the Whisper models: larger models demonstrate better generalization across diverse test sets, underscoring the importance of model capacity in tackling challenging out-of-distribution data, such as audio generated by advanced TTS models.

### 3.2.4 ON THE EFFECTIVENESS AND EFFICIENCY OF FEW-SHOT FINE-TUNING TO IMPROVE GENERALIZATION

Despite the challenges in generalizing across different datasets, we investigate whether there exist efficient solutions that can enhance models' detection performance on those challenging subsets from SONAR dataset. To this end, we conduct a case study on Wave2Vec2BERT and HuBERT, as these models perform relatively poorly on the OpenAI and SeedTTS datasets but demonstrate competitive performance on other subsets. Specifically, we generate 100 additional fake audio samples using the OpenAI TTS API and randomly select another 100 fake audio samples from the SeedTTS test set for few-shot fine-tuning. Our study yields several interesting findings.

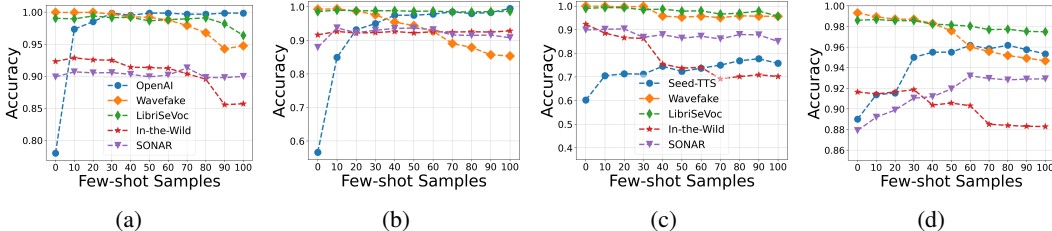

Figure 2: Performance of few-shot fine-tuning for Wave2Vec2BERT and HuBERT with a varying number of few-shot audio samples from OpenAI and Seed-TTS, respectively. (a) Fine-tune Wave2Vec2BERT on OpenAI. (b) Fine-tune HuBERT on OpenAI; (c) Fine-tune Wave2Vec2BERT on Seed-TTS; and (d) Fine-tune HuBERT on Seed-TTS.

Figures 2a and 2b present the results of fine-tuning Wave2Vec2BERT and HuBERT using varying numbers of samples from OpenAI. Before fine-tuning, Wave2Vec2BERT and HuBERT only achieve accuracies of 0.7833 and 0.5658, respectively. Notably, with only 10 shots of fake speech data, Wave2Vec2BERT reaches an accuracy of approximately 0.97, while HuBERT's accuracy increases significantly to approximately 0.85. Importantly, the models' generalization to other datasets remains unchanged, demonstrating the effectiveness and efficiency of few-shot fine-tuning. However, as the number of fine-tuning samples increases, HuBERT's test accuracy on the WaveFake test set shows a declining trend, which is also observed for Wave2Vec2BERT.

It is important to note, however, that the efficiency and effectiveness of few-shot fine-tuning may vary across different datasets. As illustrated in Figures 2c and 2d, which depict the fine-tuning results for Wave2Vec2BERT and HuBERT on Seed-TTS, the improvement in accuracy is less pronounced compared to the results on the OpenAI dataset. While the accuracy of both Wave2Vec2BERT and HuBERT does improve on Seed-TTS, the gains are not as significant as those observed for the OpenAI dataset. Additionally, the detection performance on other datasets decreases more noticeably when fine-tuning on Seed-TTS compared to OpenAI.

These findings suggest that the effectiveness of few-shot fine-tuning may depend on the specific characteristics of the dataset. Moreover, this also highlights its potential for tailored applications, such as personalized detection systems for a specific entity or individual, to enable more customized and practical applications.

### 3.2.5 How well can detection models generalize across languages?

To evaluate the cross-lingual generalization capabilities of detection models, we further conduct experiments on the MLAAD dataset (Müller et al., 2024), which encompasses 38 diverse languages. Table 7 presents the average performance metrics across all languages, with detailed language-specific results provided in Tables 10, 11, and 12 in the Appendix. Our analysis reveals several significant findings regarding model generalization across languages.

**Foundation models demonstrate remarkable cross-lingual generalization capabilities, despite being fine-tuned exclusively on English speech data**. From Table 7, it can be observed

Table 7: Generalization of different models across languages.

| Model | Accuracy | AUROC | EER (%) |
|---|---|---|---|
| LFCC-LCNN | 0.6986 | 0.7661 | 30.1447 |
| Spec.+ResNet | 0.6001 | 0.6310 | 39.9947 |
| RawNet2 | 0.4538 | 0.4379 | 54.6237 |
| RawGATST | 0.8061 | 0.8762 | 19.3921 |
| AASIST | 0.8461 | 0.9157 | 15.3868 |
| CLAP | 0.5136 | 0.5125 | 48.6395 |
| Whisper-small | 0.8276 | 0.9033 | 17.2395 |
| Whisper-large | 0.8325 | 0.9081 | 16.7474 |
| Wave2Vec2 | 0.9139 | 0.9387 | 8.5947 |
| HuBERT | 0.9320 | 0.9745 | 6.7974 |
| Wave2Vec2BERT | 0.9901 | 0.9950 | 0.9921 |

that Wave2Vec2BERT achieves exceptional performance with an accuracy of 0.9901, AUROC of 0.9950, and EER of 0.9921%. Similarly, HuBERT and Wave2Vec2 also show strong performance, with accuracies of 0.9320 and 0.9139, respectively. This robust cross-lingual generalization may be attributed to The diverse multilingual pretraining data these models are exposed to during their self-supervised learning phase. Their ability to learn language-agnostic speech representations that capture fundamental acoustic properties relevant to deepfake detection.

In contrast, traditional detection models show varying degrees of success in cross-lingual generalization. AASIST and RawGATST achieve respectable average accuracies of 0.846 and 0.806 on MLAAD, respectively. However, their performance significantly degrades on the SONAR dataset (accuracies of 0.6975 and 0.5939, as shown in Table 3a). The disparity in performance between MLAAD (containing primarily open-source TTS-generated audio) and SONAR provides crucial insights. Foundation models maintain relatively consistent performance across both datasets, while traditional detection models show significant degradation on SONAR. This observation suggests that **the primary challenges in audio deepfake detection are more closely tied to the realism and quality of synthetic audio rather than language-specific characteristics.**

## 4    DISCUSSION

**AI-synthetized audio detection methods must be evaluated on diverse and advanced benchmarks.** In our evaluation using the proposed dataset, most models perform well on standard TTS tools but suffer significant degradation when tested on the fake audios generated by the most advanced tool such as Voice Engine released by OpenAI. Therefore, we advocate for future research in audio deepfake detection to prioritize benchmarking against the latest and most advanced TTS technologies, which will lead to more robust and reliable detectors, as relying on high detection rates from outdated tools may create a false sense of generalization. Additionally, there is an urgent need to develop larger-scale training datasets comprising fake audio generated by cutting-edge TTS models to keep pace with rapid advancements in TTS technology and mitigate associated risks.

**Limitations and future work**. While our primary goal in proposing this dataset is to facilitate comprehensive evaluation, it remains relatively small in size and is primarily focused on English. A more in-depth analysis of detection performance across different languages and gender representations is crucial for a more comprehensive evaluation. These aspects are essential for future research to enhance the dataset's applicability and generalizability. For future work, we also plan to: (1) incorporate additional AI-audio detection models, including those targeting advanced audio editing techniques designed to bypass detection systems; (2) explore innovative methods to further improve generalizability; and (3) address realistic challenges and risks in deploying the proposed method in real-world scenarios, such as evaluating the robustness of models against common or adversarial corruptions. These efforts will contribute to the development of more effective strategies to combat AI-generated audio threats.

**Data license considerations.**. Since our dataset is sourced from various models, each may be subject to distinct distribution licenses and usage restrictions. Throughout the data collection process, we strictly adhered to all relevant usage policies. The dataset is made accessible either directly through the provided link or indirectly via the original sources. To account for the evolving nature of these policies, we are committed to keeping the published dataset fully compliant with the latest regulations. Additionally, we will reference the usage policies of the respective API providers to inform users about any potential restrictions.

## 5    CONCLUSION

In this paper, we presented SONAR, a framework providing a comprehensive evaluation for distinguishing state-of-the-art AI-synthesized auditory content. SONAR introduces a novel evaluation dataset sourced from 9 diverse audio synthesis platforms, including leading TTS service providers and state-of-the-art TTS models. To the best of our knowledge, SONAR is the first platform that provides uniform, comprehensive, informative, and extensible evaluation of deepfake audio detection models. Leveraging SONAR, we conducted extensive experiments to analyze the generalizability limitations of current detection methods. We found that speech foundation models demonstrate stronger generalization capabilities across datasets and languages, given their massive model size scale and pertaining data. We also suggest that the primary challenges in audio deepfake detection are more closely tied to the realism and quality of synthetic audio rather than language-specific characteristics. In addition, we further explored the potential of few-shot fine-tuning to improve generalization and demonstrated its efficiency and effectiveness. We envision that SONAR will serve as a valuable benchmark to facilitate research in AI-audio detection and highlight directions for further improvement.

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

## A APPENDIX

### A.1 RELATED WORK

**Text-to-Speech synthesis**. Human voice synthesis is a significant challenge in the field of AI. State-of-the-art TTS synthesis approaches such as VALLE (Wang et al., 2023), AudioBox (Vyas et al., 2023), VoiceBox (Le et al., 2024), NaturalSpeech3 (Wang et al., 2023), and YourTTS (Casanova et al., 2022) have demonstrated the possibility of generating high-quality, human-realistic audio with generative models trained on large datasets. Current TTS models can be classified into two primary categories: cascaded and end-to-end methods. Cascaded TTS models (Shen et al., 2018; Ren et al., 2019; Li et al., 2019) typically employ a pipeline involving an acoustic model and a vocoder utilizing mel spectrograms as intermediary representations. To address the limitations associated with vocoders, end-to-end TTS models (Kim et al., 2021; Liu et al., 2022) have been developed to jointly optimize both the acoustic model and vocoder. In practical applications, it is preferable to customize TTS systems to generate speech in any voice with limited accessible data. Consequently, there is increasing interest in zero-shot multi-speaker TTS techniques (Cooper et al., 2020; Casanova et al., 2022; Ye et al., 2024).

**AI-synthesized audio detection.** Recent advancements in AI technology have significantly enhanced the ability to generate high-quality and realistic audio, calling for an urgent need for more robust and reliable detection methods. Several datasets have been developed to support research in this area. The ASVspoof challenges (Wu et al., 2017; Wang et al., 2020; Nautsch et al., 2021; Todisco et al., 2019; Liu et al., 2023) are among the most notable, offering comprehensive datasets

that cover a variety of attack vectors, including replay attacks, voice conversion, and directly synthesized audio. These resources aim to facilitate thorough evaluations of countermeasures against various spoofing techniques. In addition, newer datasets such as WaveFake (Frank & Schönherr, 2021) and LibriSeVoc (Sun et al., 2023) provide fake audio samples generated with state-of-the-art vocoders, offering diverse distributions to enhance the development of deepfake audio detection systems. By comparison, the In-the-Wild dataset (Zang et al., 2024) targets real-world applications by collecting deepfake audios from publicly accessible sources, capturing the complexity and diversity of manipulations encountered in everyday environments. Similarly, the SingFake dataset (Zang et al., 2024) focuses on the detection of synthetic singing voices, presenting unique challenges due to the musical content and variation in vocal expressions. Müller et al. (2024) presents MLAAD dataset, which consists of fake audios created using 82 TTS models, covering 38 different languages. These datasets are crucial for developing and testing next-generation AI-synthesized audio detection systems, pushing the boundaries of what is achievable in identifying and mitigating the threats posed by advanced audio synthesis technologies.

Building upon these datasets, a significant body of research has focused on distinguishing AI-generated audio from genuine audio by designing advanced model architectures (Tak et al., 2021b; Lavrentyeva et al., 2019; Jung et al., 2022; Tak et al., 2021a) tailored for extracting different levels of representations of speech data for audio deepfake detection. Additionally, recent works (Wang & Yamagishi, 2021; Tak et al., 2022; Kawa et al., 2023) have leveraged speech foundation models for audio deepfake detection tasks. For instance, Wang & Yamagishi (2021) and Tak et al. (2022) fine-tune Wav2Vec2 (Baevski et al., 2020) models on the ASVspoof dataset, while Kawa et al. (2023) uses Whisper as a front-end to extract audio features and trains various detection models based on these features, achieving state-of-the-art detection performance on the corresponding test datasets. However, none of these models have been evaluated on audio generated by the latest text-to-speech (TTS) models, leaving a gap in understanding their effectiveness against the most recent advancements in synthetic audio generation.

Several recent work has also focused on benchmarking audio deepfake detection models, though with varying scope and limitations. Alali & Theodorakopoulos (2024) provides an overview of TTS, VC, and PF methods but lacks empirical validation, while Chen et al. (2020b) limits their evaluation to the ASVSpoof2019 dataset(Wang et al., 2020). Yan et al. (2024) constructs a bilingual dataset and evaluates several detection models, but part of their data generated by commercial APIs raises data license issues. Similar to SONAR, a concurrent work (Xie et al., 2024) collected samples from web demos of advanced TTS models but evaluated only two detection methods. Similarly, Li et al. (2024) benchmarks three foundation models on open-source TTS-generated data and studies the potential threats posed by audio perturbations.

In contrast, SONAR offers several key advantages: (1) it provides comprehensive empirical evaluation of both traditional detection models and foundation models, including those not previously explored; (2) it incorporates a diverse range of audio sources, from web demos to compliant commercial APIs, while ensuring adherence to usage policies; (3) it examines the impact of model architecture, training data, and few-shot fine-tuning on detection performance; and (4) it reveals critical insights about the trade-off between fine-tuning effectiveness and model generalization. These contributions make SONAR a more extensive and systematic benchmark for audio deepfake detection.

## A.2 BROAD IMPACTS

**Societal Risks**. The rapid advancement of AI-Generated Content (AIGC) in audio and speech poses significant societal risks as it becomes more prevalent in audio and speech generation. As our work in benchmarking AI-synthesized audio detection demonstrates, the line between AI-generated audio and human speech is increasingly blurring, making it difficult for individuals to distinguish between synthetic and authentic voices. This raises serious concerns about spreading misinformation and fabricating narratives. AI-generated speeches could be used to impersonate public figures, spread false information, or even incite unrest by delivering provocative messages that appear authentic. For example, deepfake audios of political figures can be created to falsely represent their opinions or statements, potentially influencing public perception and affecting democratic processes.

Table 8: Hyperparemeters

| config | value |
|---|---|
| optimizer | Adam |
| optimizer momentum | $\beta_1 = 0.9$, $\beta_2 = 0.999$ |
| weight decay | 1e-4 |
| epochs | 40 |
| warmup epochs | 0 |
| scheduler | cosine decay |

Moreover, these technologies could be exploited to damage reputations or cause legal issues for individuals or organizations through fake endorsements or harmful statements. It is crucial for academia and industry to develop robust detection methods and ethical guidelines to prevent misuse of this technology and to educate the public about its capabilities and associated risks.

**Positive Impacts**. On the positive side, AI-synthesized audio/speech has the potential to revolutionize content creation in various sectors, including education, entertainment, and accessibility. In education, AI-synthesized audios and speeches enables production of customized content that meets diverse learning needs and languages, improving access and inclusivity. For entertainment, they can offer novel experiences by generating dynamic dialogues in games or virtual reality, enriching user engagement and creativity.

Furthermore, AI-synthesized audios and speeches also enhances accessibility by producing speech in various languages or dialects, bridging communication gaps and making information more accessible to non-native speakers or those with liabilities. Additionally, the technology can help preserve lesser-spoken languages and dialects at risk of extinction by creating archives of AI-generated speeches and narratives.

In conclusion, while AI-synthesized audios and speeches offer exciting opportunities for content creation and accessibility, it is essential to address the ethical and societal challenges associated with its use. Collaborative efforts among researchers, developers, and policymakers are crucial to leveraging AI-synthesized audio and speech benefits responsibly while mitigating its risks, ensuring the technology serves to enhance human communication and creativity positively and responsibly.

A.3 Implementation Details

Table 8 presents the hyperparameters for training AASIST, RawNet2, RawGAT-ST, LCNN, and Spec.+ResNet. We train AASIST, RawNet2, and RawGAT-ST with a learning rate of 0.0001 and LCNN and Spec.+ResNet with a learning rate of 0.0003. The batch size for AASIST, RawNet2, RawGAT-ST, LCNN, and Spec.+ResNet are 64, 256, 32, 512, and 256, respectively. All input audios are resampled to a 16kHz sampling rate and converted into raw waveforms consisting of 64,000 samples (approximately 4 seconds). Audios longer than 4 seconds are randomly trimmed, while those shorter than 4 seconds are repeated and padded to meet the 4-second duration.

For the foundation models, two linear layers are added after the encoder's output, with the hidden layer dimension matching the dimension of the encoder's output. We fine-tune all foundation models on the Wavefake training dataset for 3 epochs using the Adam optimizer with a learning rate of 0.00001 and a weight decay of 0.0005.

For few-shot fine-tuning, models are fine-tuned for 30 epochs with a learning rate of 0.00001 and a weight decay of 0.00005.

A.4 Evaluation of the models' robustness against audio corruptions

We further evaluate the robustness of various models on the WaveFake test set. Specifically, we tested models under MP3 compression at different bitrates and varying levels of signal-to-noise ratio (SNR) for background noise. Figure 3a and Figure 3b present the results.

Consistent with our findings on generalizability, foundation models demonstrate greater robustness to these types of corruption despite not encountering them during training. In contrast, traditional

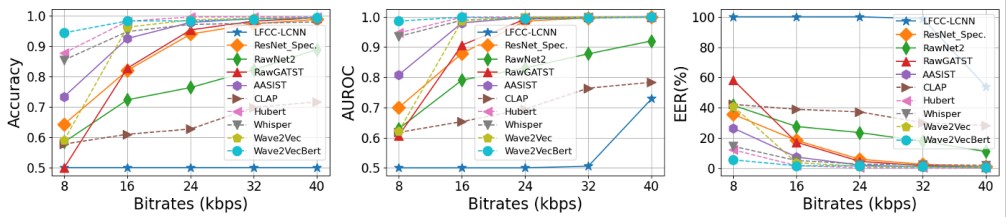

(a) Evaluation of different detection models' robustness against MP3 compression at different bitrates.

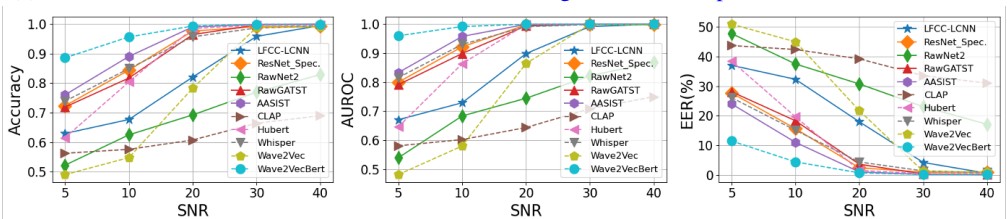

(b) Evaluation of the models' robustness against background noise at varying levels of signal-to-noise (SNR).

Figure 3: Evaluation of different detection models' robustness against speech encoding corruption (MP3) and background noise corruption.

models experience significant performance degradation in terms of accuracy and AUROC, particularly when the audios are subjected to more severe levels of corruption.

## A.5   RESULTS ON AUDIOS GENERATED BY VOICE CONVERSION

In addition to fake audios generated by various TTS models, we also evaluate the models' detection performance on audios generated by Voice Conversion (VC) technology. Specifically, we evaluate different models on the VC subset from the ASVSpoof2019 dataset (Wang et al., 2020). Table 9 presents the results. It can be observed that foundation models continue to generalize well on VC tasks, whereas traditional detection models experience a significant performance drop.

## A.6   RESULTS ON DIFFERENT TRAINING DATASET

To investigate the impact of training dataset, we further adopt ASVspoof2019 (Wang et al., 2020) training set to train/finetune different detection models/foundation models. Table 13 presents the evaluation results of models trained/finetuned on the ASVSpoof2019 dataset across different public datasets, while Table 14 presents their evaluation on the SONAR dataset. Table 15 presents the evaluation results of models trained/finetuned on the combination of ASVSpoof2019 and Wavefake dataset, while Table 16 provides their evaluation on SONAR dataset.

The results demonstrate that models only trained/finetuned on ASVSpoof2019 have worse detection performances, which is also echoed by the results in (Li et al., 2024). Compared with models only trained on ASVSpoof2019, the combination of Wavefake can further improve models' generalizability. However, models suffer from degradation in their detection performance compared with only trained/finetuned on Wavefake dataset (the results in the paper). We attribute this to the quality of the training data. The audio data in ASVSpoof2019 are generated by TTS/VC models before 2019.

We also advocate that, with the rapid development of TTS technologies, we need to adapt the training dataset distribution to higher-quality fake audios to improve the generalization of these detection models to develop better countermeasures.

Table 9: Evaluation on Voice Conversion subset from ASVSpoof2019.

| Model | Accuracy(↑) | AUROC(↑) | EER (%)(↓) |
|---|---|---|---|
| LFCC-LCNN | 0.623 | 0.667 | 37.44 |
| ResNet_Spec. | 0.660 | 0.714 | 34.05 |
| RawNet2 | 0.430 | 0.391 | 58.72 |
| RawGATST | 0.671 | 0.738 | 32.86 |
| AASIST | 0.764 | 0.841 | 23.52 |
| CLAP | 0.494 | 0.495 | 50.30 |
| Whisper-small | 0.994 | 1.000 | 0.56 |
| Whisper-large | 0.931 | 0.978 | 6.93 |
| Wave2Vec | 0.883 | 0.952 | 11.64 |
| HuBERT | 0.971 | 0.994 | 2.87 |
| Wave2VecBERT | 0.974 | 0.995 | 2.58 |

Table 10: Accuracy (↑) of different detection models across languages.

| Language | LFCC LCNN | ResNet Spec. | Raw Net2 | Raw GATST | AASIST | CLAP | Whisper small | Whisper large | Wave2Vec2 | HuBERT | Wave2Vec2 BERT |
|---|---|---|---|---|---|---|---|---|---|---|---|
| Romanian | 0.747 | 0.702 | 0.498 | 0.808 | 0.848 | 0.511 | 0.744 | 0.737 | 0.955 | 0.875 | 0.998 |
| Croatian | 0.960 | 0.823 | 0.403 | 0.973 | 0.970 | 0.424 | 0.946 | 0.973 | 0.999 | 0.977 | 0.999 |
| Dutch | 0.675 | 0.558 | 0.524 | 0.769 | 0.844 | 0.520 | 0.816 | 0.789 | 0.780 | 0.944 | 0.993 |
| Latvian | 0.579 | 0.761 | 0.417 | 0.979 | 0.976 | 0.537 | 0.906 | 0.921 | 1.000 | 0.981 | 1.000 |
| Ukrainian | 0.651 | 0.624 | 0.445 | 0.800 | 0.822 | 0.491 | 0.752 | 0.750 | 0.982 | 0.935 | 0.998 |
| Irish | 0.767 | 0.584 | 0.539 | 0.858 | 0.894 | 0.456 | 0.896 | 0.900 | 0.975 | 0.941 | 0.997 |
| Polish | 0.644 | 0.581 | 0.528 | 0.825 | 0.871 | 0.523 | 0.813 | 0.822 | 0.941 | 0.941 | 0.998 |
| Lithuanian | 0.780 | 0.723 | 0.582 | 0.944 | 0.970 | 0.468 | 0.756 | 0.732 | 1.000 | 0.956 | 0.999 |
| Chinese | 0.537 | 0.546 | 0.438 | 0.664 | 0.736 | 0.405 | 0.794 | 0.779 | 0.814 | 0.943 | 0.998 |
| Greek | 0.769 | 0.508 | 0.482 | 0.862 | 0.879 | 0.421 | 0.850 | 0.879 | 0.981 | 0.939 | 0.995 |
| German | 0.691 | 0.695 | 0.424 | 0.804 | 0.843 | 0.640 | 0.773 | 0.775 | 0.915 | 0.914 | 0.995 |
| Turkish | 0.758 | 0.578 | 0.432 | 0.779 | 0.820 | 0.562 | 0.867 | 0.840 | 0.809 | 0.952 | 0.996 |
| Russian | 0.631 | 0.540 | 0.386 | 0.646 | 0.728 | 0.567 | 0.719 | 0.701 | 0.941 | 0.915 | 0.992 |
| Arabic | 0.748 | 0.548 | 0.510 | 0.758 | 0.832 | 0.534 | 0.856 | 0.845 | 0.815 | 0.958 | 0.998 |
| Spanish | 0.606 | 0.548 | 0.440 | 0.772 | 0.860 | 0.542 | 0.687 | 0.690 | 0.930 | 0.915 | 0.995 |
| Estonian | 0.712 | 0.579 | 0.445 | 0.875 | 0.904 | 0.388 | 0.891 | 0.900 | 0.989 | 0.932 | 0.998 |
| Thai | 0.643 | 0.381 | 0.473 | 0.807 | 0.851 | 0.464 | 0.805 | 0.871 | 0.983 | 0.929 | 0.996 |
| Bulgarian | 0.705 | 0.583 | 0.462 | 0.849 | 0.892 | 0.505 | 0.867 | 0.877 | 0.994 | 0.936 | 0.998 |
| Vietnamese | 0.764 | 0.582 | 0.518 | 0.760 | 0.826 | 0.545 | 0.935 | 0.937 | 0.959 | 0.985 | 1.000 |
| Maltese | 0.777 | 0.563 | 0.482 | 0.871 | 0.904 | 0.411 | 0.896 | 0.910 | 0.990 | 0.937 | 0.997 |
| Persian | 0.480 | 0.621 | 0.327 | 0.819 | 0.713 | 0.712 | 0.899 | 0.914 | 0.991 | 0.939 | 0.998 |
| English | 0.547 | 0.663 | 0.478 | 0.683 | 0.740 | 0.445 | 0.711 | 0.675 | 0.599 | 0.891 | 0.776 |
| Turkman | 0.697 | 0.399 | 0.454 | 0.815 | 0.854 | 0.520 | 0.851 | 0.880 | 0.990 | 0.924 | 0.998 |
| Hungarian | 0.657 | 0.584 | 0.440 | 0.719 | 0.791 | 0.507 | 0.764 | 0.767 | 0.807 | 0.896 | 0.994 |
| Swedish | 0.709 | 0.593 | 0.337 | 0.827 | 0.904 | 0.451 | 0.889 | 0.890 | 0.983 | 0.934 | 0.997 |
| Japanese | 0.707 | 0.518 | 0.500 | 0.764 | 0.826 | 0.462 | 0.857 | 0.840 | 0.808 | 0.940 | 0.996 |
| Bengali | 0.589 | 0.594 | 0.523 | 0.606 | 0.632 | 0.518 | 0.556 | 0.640 | 0.690 | 0.729 | 0.958 |
| Italian | 0.514 | 0.565 | 0.488 | 0.806 | 0.843 | 0.508 | 0.874 | 0.874 | 0.968 | 0.943 | 0.998 |
| Finnish | 0.691 | 0.583 | 0.450 | 0.735 | 0.780 | 0.545 | 0.767 | 0.744 | 0.880 | 0.931 | 0.996 |
| Hindi | 0.727 | 0.578 | 0.491 | 0.768 | 0.838 | 0.528 | 0.857 | 0.877 | 0.817 | 0.957 | 0.998 |
| Swahili | 0.820 | 0.746 | 0.558 | 0.895 | 0.876 | 0.759 | 0.807 | 0.841 | 0.982 | 0.914 | 0.996 |
| Slovak | 0.724 | 0.605 | 0.345 | 0.886 | 0.907 | 0.453 | 0.893 | 0.898 | 0.994 | 0.938 | 0.997 |
| Danish | 0.791 | 0.626 | 0.352 | 0.886 | 0.907 | 0.593 | 0.897 | 0.912 | 0.989 | 0.938 | 0.998 |
| French | 0.652 | 0.575 | 0.400 | 0.796 | 0.838 | 0.598 | 0.764 | 0.764 | 0.971 | 0.916 | 0.996 |
| Portuguese | 0.730 | 0.599 | 0.467 | 0.749 | 0.807 | 0.562 | 0.844 | 0.841 | 0.788 | 0.957 | 0.996 |
| Korean | 0.748 | 0.546 | 0.464 | 0.761 | 0.836 | 0.536 | 0.866 | 0.888 | 0.882 | 0.953 | 0.998 |
| Slovenian | 0.874 | 0.765 | 0.291 | 0.951 | 0.944 | 0.408 | 0.917 | 0.927 | 0.995 | 0.965 | 0.996 |
| Czech | 0.744 | 0.635 | 0.450 | 0.762 | 0.847 | 0.498 | 0.867 | 0.836 | 0.847 | 0.947 | 0.997 |
| Average | 0.699 | 0.600 | 0.454 | 0.806 | 0.846 | 0.514 | 0.828 | 0.833 | 0.914 | 0.932 | 0.990 |

Table 11: AUROC (↑) of different detection models across languages.

| Language | LFCC LCNN | ResNet Spec. | Raw Net2 | Raw GATST | AASIST | CLAP | Whisper small | Whisper large | Wave2Vec | HuBERT | Wave2Vec BERT |
|---|---|---|---|---|---|---|---|---|---|---|---|
| Romanian | 0.836 | 0.750 | 0.545 | 0.902 | 0.937 | 0.490 | 0.835 | 0.839 | 0.969 | 0.942 | 1.000 |
| Croatian | 0.994 | 0.892 | 0.340 | 0.998 | 0.997 | 0.377 | 0.987 | 0.994 | 1.000 | 0.997 | 1.000 |
| Dutch | 0.747 | 0.572 | 0.518 | 0.853 | 0.920 | 0.533 | 0.901 | 0.875 | 0.841 | 0.985 | 0.999 |
| Latvian | 0.627 | 0.818 | 0.372 | 0.999 | 0.998 | 0.521 | 0.970 | 0.970 | 1.000 | 0.996 | 1.000 |
| Ukrainian | 0.713 | 0.664 | 0.450 | 0.885 | 0.895 | 0.474 | 0.846 | 0.852 | 0.995 | 0.977 | 1.000 |
| Irish | 0.839 | 0.613 | 0.573 | 0.929 | 0.958 | 0.434 | 0.958 | 0.966 | 0.984 | 0.983 | 1.000 |
| Polish | 0.709 | 0.600 | 0.541 | 0.899 | 0.938 | 0.533 | 0.905 | 0.915 | 0.961 | 0.982 | 1.000 |
| Lithuanian | 0.869 | 0.773 | 0.565 | 0.991 | 0.997 | 0.430 | 0.842 | 0.812 | 1.000 | 0.991 | 1.000 |
| Chinese | 0.525 | 0.599 | 0.409 | 0.703 | 0.764 | 0.378 | 0.874 | 0.867 | 0.866 | 0.983 | 1.000 |
| Greek | 0.856 | 0.488 | 0.468 | 0.929 | 0.949 | 0.400 | 0.931 | 0.945 | 0.992 | 0.979 | 0.999 |
| German | 0.781 | 0.749 | 0.394 | 0.874 | 0.911 | 0.676 | 0.864 | 0.872 | 0.942 | 0.970 | 1.000 |
| Turkish | 0.844 | 0.637 | 0.388 | 0.861 | 0.911 | 0.596 | 0.943 | 0.919 | 0.868 | 0.989 | 1.000 |
| Russian | 0.664 | 0.570 | 0.334 | 0.713 | 0.809 | 0.571 | 0.807 | 0.798 | 0.957 | 0.968 | 1.000 |
| Arabic | 0.840 | 0.573 | 0.528 | 0.841 | 0.911 | 0.555 | 0.932 | 0.915 | 0.869 | 0.988 | 1.000 |
| Spanish | 0.656 | 0.586 | 0.415 | 0.854 | 0.934 | 0.535 | 0.788 | 0.787 | 0.948 | 0.962 | 0.999 |
| Estonian | 0.787 | 0.601 | 0.405 | 0.942 | 0.967 | 0.365 | 0.955 | 0.961 | 0.997 | 0.983 | 1.000 |
| Thai | 0.714 | 0.337 | 0.448 | 0.880 | 0.927 | 0.425 | 0.902 | 0.950 | 0.992 | 0.969 | 1.000 |
| Bulgarian | 0.772 | 0.579 | 0.450 | 0.925 | 0.957 | 0.493 | 0.942 | 0.949 | 1.000 | 0.986 | 1.000 |
| Vietnamese | 0.855 | 0.621 | 0.539 | 0.849 | 0.909 | 0.574 | 0.984 | 0.987 | 0.974 | 0.999 | 1.000 |
| Maltese | 0.859 | 0.575 | 0.490 | 0.941 | 0.966 | 0.397 | 0.959 | 0.978 | 0.998 | 0.986 | 1.000 |
| Persian | 0.484 | 0.706 | 0.223 | 0.880 | 0.811 | 0.740 | 0.958 | 0.975 | 0.997 | 0.964 | 1.000 |
| English | 0.573 | 0.725 | 0.472 | 0.735 | 0.810 | 0.418 | 0.786 | 0.763 | 0.650 | 0.933 | 0.825 |
| Turkman | 0.775 | 0.337 | 0.442 | 0.886 | 0.932 | 0.510 | 0.927 | 0.954 | 0.998 | 0.972 | 1.000 |
| Hungarian | 0.720 | 0.628 | 0.407 | 0.828 | 0.893 | 0.516 | 0.846 | 0.862 | 0.853 | 0.955 | 1.000 |
| Swedish | 0.773 | 0.593 | 0.330 | 0.896 | 0.956 | 0.434 | 0.955 | 0.955 | 0.992 | 0.982 | 1.000 |
| Japanese | 0.797 | 0.532 | 0.520 | 0.855 | 0.917 | 0.445 | 0.938 | 0.923 | 0.861 | 0.985 | 1.000 |
| Bengali | 0.643 | 0.637 | 0.522 | 0.630 | 0.681 | 0.553 | 0.632 | 0.741 | 0.770 | 0.820 | 0.991 |
| Italian | 0.542 | 0.598 | 0.470 | 0.891 | 0.917 | 0.503 | 0.949 | 0.949 | 0.978 | 0.981 | 1.000 |
| Finnish | 0.774 | 0.599 | 0.400 | 0.829 | 0.894 | 0.601 | 0.857 | 0.824 | 0.914 | 0.977 | 1.000 |
| Hindi | 0.807 | 0.594 | 0.504 | 0.847 | 0.920 | 0.544 | 0.939 | 0.953 | 0.875 | 0.989 | 1.000 |
| Swahili | 0.908 | 0.805 | 0.620 | 0.955 | 0.937 | 0.822 | 0.883 | 0.904 | 0.993 | 0.966 | 1.000 |
| Slovak | 0.811 | 0.617 | 0.309 | 0.947 | 0.967 | 0.436 | 0.961 | 0.971 | 0.999 | 0.985 | 1.000 |
| Danish | 0.873 | 0.667 | 0.307 | 0.955 | 0.954 | 0.593 | 0.961 | 0.977 | 0.997 | 0.984 | 1.000 |
| French | 0.713 | 0.615 | 0.361 | 0.884 | 0.916 | 0.638 | 0.835 | 0.849 | 0.978 | 0.967 | 0.999 |
| Portuguese | 0.821 | 0.650 | 0.460 | 0.831 | 0.899 | 0.560 | 0.924 | 0.904 | 0.851 | 0.989 | 1.000 |
| Korean | 0.835 | 0.557 | 0.469 | 0.843 | 0.920 | 0.543 | 0.941 | 0.955 | 0.915 | 0.989 | 1.000 |
| Slovenian | 0.948 | 0.846 | 0.224 | 0.990 | 0.992 | 0.354 | 0.972 | 0.977 | 1.000 | 0.991 | 1.000 |
| Czech | 0.828 | 0.676 | 0.428 | 0.849 | 0.929 | 0.505 | 0.940 | 0.919 | 0.899 | 0.990 | 1.000 |
| Average | 0.766 | 0.631 | 0.438 | 0.876 | 0.916 | 0.513 | 0.903 | 0.908 | 0.939 | 0.975 | 0.995 |

Table 12: EER (%) (↓) of different detection models across languages.

| Language | LFCC LCNN | ResNet Spec. | Raw Net2 | Raw GATST | AASIST | CLAP | Whisper small | Whisper large | Wave2Vec | HuBERT | Wave2Vec BERT |
|---|---|---|---|---|---|---|---|---|---|---|---|
| Romanian | 25.30 | 29.80 | 50.20 | 19.20 | 15.20 | 48.90 | 25.60 | 26.30 | 4.50 | 1.25 | 0.20 |
| Croatian | 4.00 | 17.70 | 59.70 | 2.70 | 3.00 | 57.60 | 5.40 | 2.70 | 0.10 | 2.30 | 0.10 |
| Dutch | 32.50 | 44.20 | 47.60 | 23.10 | 15.60 | 48.00 | 18.40 | 21.10 | 21.90 | 5.60 | 0.70 |
| Latvian | 42.10 | 23.90 | 58.30 | 2.10 | 2.40 | 46.30 | 9.40 | 7.90 | 0.00 | 1.90 | 0.00 |
| Ukrainian | 34.90 | 37.60 | 55.50 | 20.00 | 17.80 | 50.90 | 24.80 | 25.00 | 1.80 | 6.50 | 0.20 |
| Irish | 23.30 | 41.60 | 46.10 | 14.20 | 10.60 | 54.40 | 10.40 | 10.00 | 2.50 | 5.90 | 0.30 |
| Polish | 35.60 | 41.90 | 47.20 | 17.50 | 12.90 | 47.70 | 18.70 | 17.80 | 5.90 | 5.90 | 0.20 |
| Lithuanian | 22.00 | 27.70 | 41.80 | 5.60 | 3.00 | 53.20 | 24.40 | 26.80 | 0.00 | 4.40 | 0.10 |
| Chinese | 46.30 | 45.40 | 56.20 | 33.60 | 26.40 | 59.50 | 20.60 | 22.10 | 18.60 | 5.70 | 0.20 |
| Greek | 23.10 | 49.20 | 51.80 | 13.80 | 12.10 | 57.90 | 15.00 | 12.10 | 1.90 | 6.10 | 0.50 |
| German | 30.90 | 30.50 | 57.60 | 19.60 | 15.70 | 36.00 | 22.70 | 22.50 | 8.50 | 8.60 | 0.50 |
| Turkish | 24.20 | 42.20 | 56.80 | 22.10 | 18.00 | 43.80 | 13.30 | 16.00 | 19.10 | 4.80 | 0.40 |
| Russian | 36.90 | 46.00 | 61.40 | 35.40 | 27.20 | 43.30 | 28.10 | 29.90 | 5.90 | 8.50 | 0.80 |
| Arabic | 25.20 | 45.20 | 49.00 | 24.20 | 16.80 | 46.60 | 14.40 | 15.50 | 18.50 | 4.20 | 0.20 |
| Spanish | 39.40 | 45.20 | 56.00 | 22.80 | 14.00 | 45.80 | 31.30 | 31.00 | 7.00 | 8.50 | 0.50 |
| Estonian | 28.80 | 42.10 | 55.50 | 12.50 | 9.60 | 61.20 | 10.90 | 10.00 | 1.10 | 6.80 | 0.20 |
| Thai | 35.70 | 61.90 | 52.70 | 19.30 | 14.90 | 53.60 | 19.50 | 12.90 | 1.70 | 7.10 | 0.40 |
| Bulgarian | 29.50 | 41.70 | 53.80 | 15.10 | 10.80 | 49.50 | 13.30 | 12.30 | 0.60 | 6.40 | 0.20 |
| Vietnamese | 23.60 | 41.80 | 48.20 | 24.00 | 17.40 | 45.50 | 6.50 | 6.30 | 4.10 | 1.50 | 0.00 |
| Maltese | 22.30 | 43.70 | 51.80 | 12.90 | 9.60 | 58.90 | 10.40 | 9.00 | 1.00 | 6.30 | 0.30 |
| Persian | 52.00 | 37.90 | 67.30 | 18.10 | 28.70 | 28.80 | 10.10 | 8.60 | 0.90 | 6.10 | 0.20 |
| English | 45.30 | 33.70 | 52.20 | 31.70 | 26.00 | 55.50 | 28.90 | 32.50 | 40.10 | 10.90 | 2.24 |
| Turkman | 30.30 | 60.10 | 54.60 | 18.50 | 14.60 | 48.00 | 14.90 | 12.00 | 1.00 | 7.60 | 0.20 |
| Hungarian | 34.30 | 41.60 | 56.00 | 28.10 | 20.90 | 49.30 | 23.60 | 23.30 | 19.30 | 10.40 | 0.60 |
| Swedish | 29.10 | 40.70 | 66.30 | 17.30 | 9.60 | 54.90 | 11.10 | 11.00 | 1.70 | 6.60 | 0.30 |
| Japanese | 29.30 | 48.20 | 50.00 | 23.60 | 17.40 | 53.80 | 14.30 | 16.00 | 19.20 | 6.00 | 0.40 |
| Bengali | 41.10 | 40.60 | 47.70 | 39.40 | 36.80 | 48.20 | 44.40 | 36.00 | 30.90 | 27.10 | 4.20 |
| Italian | 48.60 | 43.50 | 51.20 | 19.40 | 15.70 | 49.20 | 12.60 | 12.60 | 3.20 | 5.70 | 0.20 |
| Finnish | 30.90 | 41.70 | 55.00 | 26.50 | 22.00 | 45.50 | 23.30 | 25.60 | 12.00 | 6.90 | 0.40 |
| Hindi | 27.30 | 42.20 | 50.90 | 23.20 | 16.20 | 47.20 | 14.30 | 12.30 | 18.30 | 4.30 | 0.20 |
| Swahili | 18.00 | 25.40 | 44.20 | 10.50 | 12.40 | 24.10 | 19.30 | 15.90 | 1.80 | 8.60 | 0.40 |
| Slovak | 27.60 | 39.50 | 65.50 | 11.40 | 9.30 | 54.70 | 10.70 | 10.20 | 0.60 | 6.20 | 0.30 |
| Danish | 20.90 | 37.40 | 64.80 | 11.40 | 9.30 | 40.70 | 10.30 | 8.80 | 1.10 | 6.20 | 0.20 |
| French | 34.80 | 42.50 | 60.00 | 20.40 | 16.20 | 40.20 | 23.60 | 23.60 | 2.90 | 8.40 | 0.40 |
| Portuguese | 27.00 | 40.10 | 53.30 | 25.10 | 19.30 | 43.80 | 15.60 | 15.90 | 21.30 | 4.30 | 0.40 |
| Korean | 25.20 | 45.40 | 53.60 | 23.90 | 16.40 | 46.40 | 13.40 | 11.20 | 11.80 | 4.70 | 0.20 |
| Slovenian | 12.60 | 23.50 | 70.90 | 4.90 | 5.60 | 59.20 | 8.30 | 7.30 | 0.50 | 3.50 | 0.40 |
| Czech | 25.60 | 36.50 | 55.00 | 23.80 | 15.30 | 50.20 | 13.30 | 16.40 | 15.30 | 5.30 | 0.30 |
| Average | 30.14 | 40.00 | 54.62 | 19.39 | 15.39 | 48.64 | 17.24 | 16.75 | 8.59 | 6.80 | 0.99 |

Table 13: Generalization across existing audio deepfake datasets. All models are trained/finetuned on the ASVSpoof2019 training set.

| Model | ASVSpoof2019 | | | Wavefake | | | Libri | | | In the wild | | |
|---|---|---|---|---|---|---|---|---|---|---|---|---|
| | Acc | AUROC | EER(%) | Acc | AUROC | EER(%) | Acc | AUROC | EER(%) | Acc | AUROC | EER(%) |
| LFCC-LCNN | 0.9474 | 0.9861 | 5.2620 | 0.5660 | 0.5975 | 43.3970 | 0.6225 | 0.6780 | 37.7510 | 0.4803 | 0.4635 | 51.9719 |
| ResNet_Spec. | 0.8831 | 0.9447 | 11.6930 | 0.4798 | 0.4701 | 52.0230 | 0.5150 | 0.5122 | 48.5040 | 0.4091 | 0.3968 | 59.0890 |
| RawNet2 | 0.8697 | 0.9358 | 13.0250 | 0.5218 | 0.5332 | 47.8200 | 0.5437 | 0.5653 | 45.6267 | 0.5070 | 0.5084 | 49.2980 |
| RawGATST | 0.9618 | 0.9924 | 3.8210 | 0.5359 | 0.5592 | 46.4120 | 0.6058 | 0.6562 | 39.4170 | 0.6352 | 0.6912 | 36.4760 |
| AASIST | 0.9523 | 0.9891 | 0.9834 | 0.5252 | 0.5371 | 47.4810 | 0.5854 | 0.6182 | 41.4620 | 0.5997 | 0.6284 | 40.0300 |
| CLAP | 0.9169 | 0.9775 | 8.3070 | 0.5920 | 0.6351 | 40.8015 | 0.5717 | 0.6031 | 42.8247 | 0.7042 | 0.7837 | 29.5785 |
| Whisper-small | 0.9914 | 0.9995 | 0.8566 | 0.7786 | 0.8570 | 22.1370 | 0.8175 | 0.8947 | 18.2510 | 0.8078 | 0.8872 | 19.2200 |
| Whisper-large | 0.9923 | 0.9996 | 0.7322 | 0.7886 | 0.8630 | 21.3380 | 0.8347 | 0.9128 | 15.6820 | 0.8263 | 0.9006 | 16.6520 |
| Wave2Vec | 0.9770 | 0.9945 | 2.2978 | 0.7115 | 0.7735 | 28.8550 | 0.8012 | 0.8802 | 19.8788 | 0.7910 | 0.8543 | 20.8954 |
| HuBERT | 0.9788 | 0.9946 | 2.1210 | 0.6424 | 0.6883 | 35.7634 | 0.7993 | 0.8812 | 20.0682 | 0.7288 | 0.7868 | 27.1158 |
| Wave2VecBERT | 0.9958 | 0.9998 | 0.4215 | 0.8149 | 0.9055 | 18.5114 | 0.9379 | 98.025 | 6.2100 | 0.9366 | 0.9778 | 6.3389 |

Table 14: Evaluation on SONAR dataset. Models are only trained/finetuned on ASVSpoof2019 training set.

(a) Accuracy (↑).

| Model | PromptTTS2 | NaturalSpeech3 | VALL-E | VoiceBox | FlashSpeech | AudioGen | xTTS | Seed-TTS | OpenAI | Average |
|---|---|---|---|---|---|---|---|---|---|---|
| LFCC-LCNN | 0.5200 | 0.5625 | 0.5263 | 0.4135 | 0.5508 | 0.8900 | 0.6317 | 0.5333 | 0.7367 | 0.5960 |
| ResNet_Spec. | 0.5200 | 0.6875 | 0.5263 | 0.4327 | 0.6102 | 0.8900 | 0.4933 | 0.5083 | 0.5233 | 0.5770 |
| RawNet2 | 0.5600 | 0.5625 | 0.6000 | 0.4519 | 0.5339 | 0.5200 | 0.5517 | 0.4933 | 0.6333 | 0.5450 |
| RawGATST | 0.6400 | 0.5625 | 0.5684 | 0.4615 | 0.5000 | 0.9500 | 0.7533 | 0.5450 | 0.6183 | 0.6220 |
| AASIST | 0.6800 | 0.7188 | 0.6632 | 0.3462 | 0.5508 | 0.8200 | 0.5133 | 0.6250 | 0.7033 | 0.6250 |
| CLAP | 0.4800 | 0.5938 | 0.7474 | 0.5000 | 0.3475 | 0.3300 | 0.7100 | 0.5400 | 0.6500 | 0.5440 |
| Whisper-small | 0.9200 | 0.7812 | 0.7158 | 0.6058 | 0.4831 | 0.6100 | 0.7183 | 0.6383 | 0.4150 | 0.6540 |
| Whisper-large | 0.9300 | 0.7942 | 0.7635 | 0.6221 | 0.5368 | 0.7280 | 0.7546 | 0.6127 | 0.4430 | 0.6870 |
| Wave2Vec | 0.7600 | 0.7188 | 0.6842 | 0.8077 | 0.5508 | 0.9900 | 0.6483 | 0.8633 | 0.6633 | 0.7430 |
| HuBERT | 0.8800 | 0.7812 | 0.6842 | 0.8077 | 0.6102 | 1.0000 | 0.6117 | 0.8300 | 0.4500 | 0.7390 |
| Wave2VecBERT | 1.0000 | 0.8750 | 0.8316 | 0.8077 | 0.5763 | 0.9500 | 0.9317 | 0.6400 | 0.4517 | 0.7850 |

(b) AUROC (↑).

| Model | PromptTTS2 | NaturalSpeech3 | VALL-E | VoiceBox | FlashSpeech | AudioGen | xTTS | Seed-TTS | OpenAI | Average |
|---|---|---|---|---|---|---|---|---|---|---|
| LFCC-LCNN | 0.5552 | 0.6016 | 0.5491 | 0.4054 | 0.5184 | 0.9438 | 0.6841 | 0.5587 | 0.8049 | 0.6250 |
| ResNet_Spec. | 0.5472 | 0.6973 | 0.5436 | 0.4555 | 0.6436 | 0.9308 | 0.4735 | 0.5429 | 0.5472 | 0.5980 |
| RawNet2 | 0.6144 | 0.6406 | 0.5833 | 0.4384 | 0.5214 | 0.5526 | 0.5842 | 0.5049 | 0.6611 | 0.5670 |
| RawGATST | 0.6544 | 0.6191 | 0.5804 | 0.4296 | 0.4898 | 0.9641 | 0.8140 | 0.5813 | 0.6588 | 0.6440 |
| AASIST | 0.7584 | 0.7256 | 0.7165 | 0.2999 | 0.5985 | 0.8491 | 0.5189 | 0.6473 | 0.7671 | 0.6530 |
| CLAP | 0.4912 | 0.6895 | 0.8120 | 0.5252 | 0.2826 | 0.3074 | 0.7701 | 0.5657 | 0.7277 | 0.5750 |
| Whisper-small | 0.9520 | 0.8896 | 0.7420 | 0.6630 | 0.4844 | 0.6710 | 0.7812 | 0.7056 | 0.3924 | 0.6980 |
| Whisper-large | 0.9628 | 0.9011 | 0.8127 | 0.6938 | 0.5763 | 1.0000 | 0.7035 | 0.6765 | 0.4528 | 0.7530 |
| Wave2Vec | 0.8304 | 0.8037 | 0.7485 | 0.8805 | 0.5826 | 0.9999 | 0.6766 | 0.9404 | 0.7070 | 0.7970 |
| HuBERT | 0.9424 | 0.9043 | 0.7485 | 0.8992 | 0.6431 | 1.0000 | 0.6759 | 0.9080 | 0.4664 | 0.7990 |
| Wave2VecBERT | 1.0000 | 0.9248 | 0.8693 | 0.8828 | 0.6319 | 0.9796 | 0.9812 | 0.6825 | 0.4252 | 0.8200 |

(c) EER(%) (↓).

| Model | PromptTTS2 | NaturalSpeech3 | VALL-E | VoiceBox | FlashSpeech | AudioGen | xTTS | Seed-TTS | OpenAI | Average |
|---|---|---|---|---|---|---|---|---|---|---|
| LFCC-LCNN | 48.0000 | 43.7500 | 47.3684 | 58.6538 | 44.9153 | 11.0000 | 36.8333 | 46.6667 | 23.3333 | 40.0580 |
| ResNet_Spec. | 48.0000 | 31.2500 | 47.3684 | 56.7308 | 38.9831 | 11.0000 | 50.6667 | 49.1667 | 47.6667 | 42.3150 |
| RawNet2 | 44.0000 | 43.7500 | 40.0000 | 54.8077 | 46.6102 | 48.0000 | 44.8333 | 50.6667 | 36.6667 | 45.4820 |
| RawGATST | 36.0000 | 43.7500 | 43.1579 | 53.8462 | 50.0000 | 5.0000 | 24.6667 | 45.5000 | 38.1667 | 37.7880 |
| AASIST | 32.0000 | 28.1250 | 33.6842 | 65.3846 | 44.9153 | 18.0000 | 48.6667 | 37.5000 | 29.6667 | 37.5490 |
| CLAP | 52.0000 | 40.6250 | 25.2632 | 50.0000 | 65.2542 | 67.0000 | 29.0000 | 46.0000 | 35.0000 | 45.5710 |
| Whisper-small | 8.0000 | 21.8750 | 28.4211 | 39.4231 | 51.6949 | 39.0000 | 28.1667 | 36.4667 | 58.5000 | 34.6160 |
| Whisper-large | 7.0000 | 20.5830 | 23.6590 | 37.7920 | 46.3280 | 27.2210 | 24.5450 | 38.7330 | 55.7360 | 31.2890 |
| Wave2Vec | 24.0000 | 28.1250 | 31.5789 | 19.2308 | 44.9153 | 1.0000 | 35.1667 | 13.6667 | 33.6667 | 25.7060 |
| HuBERT | 12.0000 | 21.8750 | 31.5789 | 19.2308 | 38.9831 | 0.0000 | 38.8333 | 17.0000 | 55.0000 | 26.0560 |
| Wave2VecBERT | 0.0000 | 12.5000 | 16.8421 | 19.2308 | 42.3729 | 5.0000 | 6.8333 | 36.0000 | 54.8333 | 21.5120 |

Table 15: Generalization across existing audio deepfake datasets. All models are trained/finetuned on the combination of ASVSpoof2019 and Wavefake training set.

| Model | ASVSpoof2019 | | | Wavefake | | | Libri | | | In the wild | | |
|---|---|---|---|---|---|---|---|---|---|---|---|---|
| | Acc | AUROC | EER(%) | Acc | AUROC | EER(%) | Acc | AUROC | EER(%) | Acc | AUROC | EER(%) |
| LFCC-LCNN | 0.9414 | 0.9841 | 5.8600 | 0.5000 | 0.7474 | 37.7480 | 0.6797 | 0.7474 | 32.0330 | 0.5000 | 0.7474 | 37.7480 |
| ResNet_Spec. | 0.8942 | 0.9506 | 10.5778 | 0.5691 | 0.6089 | 43.2443 | 0.5687 | 0.5842 | 43.1280 | 0.4135 | 0.3967 | 58.6490 |
| RawNet2 | 0.8851 | 0.9482 | 11.4890 | 0.5811 | 0.6324 | 42.1370 | 0.6134 | 0.6606 | 38.6590 | 0.5267 | 0.5362 | 47.3340 |
| RawGATST | 0.9616 | 0.9918 | 3.8340 | 0.5256 | 0.5447 | 47.4430 | 0.6024 | 0.6409 | 39.7577 | 0.6315 | 0.6771 | 36.8480 |
| AASIST | 0.9615 | 0.9921 | 3.8480 | 0.5385 | 0.5527 | 46.1450 | 0.6073 | 0.6516 | 39.2650 | 0.6418 | 0.7010 | 35.8160 |
| CLAP | 0.9065 | 0.9675 | 9.3540 | 0.6912 | 0.7761 | 30.7630 | 0.6838 | 0.7494 | 31.6170 | 0.6206 | 0.6682 | 37.9400 |
| Whisper-small | 0.9812 | 0.9986 | 1.8760 | 0.9340 | 0.9821 | 6.6030 | 0.9175 | 0.9763 | 8.2540 | 0.8626 | 0.9380 | 13.7440 |
| Whisper-large | 0.9889 | 0.9994 | 1.2580 | 0.9422 | 0.9953 | 5.2270 | 0.9369 | 0.9892 | 6.3220 | 0.8823 | 0.9540 | 11.6520 |
| Wave2Vec | 0.9772 | 0.9955 | 2.2840 | 0.6336 | 0.6857 | 36.6410 | 0.9667 | 0.9941 | 3.3332 | 0.8512 | 0.9258 | 14.8780 |
| HuBERT | 0.9917 | 0.9995 | 0.8294 | 0.8996 | 0.9599 | 9.8092 | 0.9864 | 0.9986 | 1.3631 | 0.9345 | 0.9821 | 6.5504 |
| Wave2VecBERT | 0.9562 | 0.9869 | 4.3640 | 0.6815 | 0.7488 | 31.8700 | 0.9814 | 0.9910 | 1.8554 | 0.9308 | 0.9622 | 6.9228 |

Table 16: Evaluation on SONAR dataset. Models are trained/finetuned on the combination of ASVSpoof2019 and Wavefake training set.

(a) Accuracy (↑).

| Model | PromptTTS2 | NaturalSpeech3 | VALL-E | VoiceBox | FlashSpeech | AudioGen | xTTS | Seed-TTS | OpenAI | Average |
|---|---|---|---|---|---|---|---|---|---|---|
| LFCC-LCNN | 0.5600 | 0.7500 | 0.7895 | 0.5865 | 0.7627 | 0.8300 | 0.7167 | 0.4167 | 0.6233 | 0.6710 |
| ResNet_Spec. | 0.6800 | 0.6875 | 0.6105 | 0.5769 | 0.6864 | 0.9100 | 0.5617 | 0.5667 | 0.6500 | 0.6590 |
| RawNet2 | 0.6000 | 0.5625 | 0.6421 | 0.4327 | 0.6017 | 0.8700 | 0.6683 | 0.4583 | 0.6633 | 0.6110 |
| RawGATST | 0.7200 | 0.6522 | 0.7324 | 0.4458 | 0.5822 | 0.9800 | 0.7782 | 0.5211 | 0.6745 | 0.6760 |
| AASIST | 0.7300 | 0.6413 | 0.7667 | 0.5023 | 0.6136 | 0.8900 | 0.6852 | 0.6285 | 0.6322 | 0.6770 |
| CLAP | 0.6400 | 0.8125 | 0.7579 | 0.3365 | 0.4661 | 0.7600 | 0.6317 | 0.3400 | 0.7567 | 0.6110 |
| Whisper-small | 0.9600 | 0.7500 | 0.7474 | 0.7404 | 0.4407 | 0.9000 | 0.8050 | 0.6300 | 0.3533 | 0.7030 |
| Whisper-large | 0.9800 | 0.7700 | 0.7893 | 0.7822 | 0.5883 | 0.9600 | 0.8411 | 0.6216 | 0.4688 | 0.7560 |
| Wave2Vec | 0.9200 | 0.7188 | 0.7684 | 0.9038 | 0.6059 | 0.9400 | 0.6558 | 0.8742 | 0.6642 | 0.7830 |
| HuBERT | 1.0000 | 0.8125 | 0.8842 | 0.9615 | 0.7966 | 1.0000 | 0.8417 | 0.9067 | 0.7150 | 0.8800 |
| Wave2VecBERT | 1.0000 | 0.6250 | 0.7789 | 0.8846 | 0.6017 | 0.9800 | 0.9483 | 0.5558 | 0.4917 | 0.7630 |

(b) AUROC (↑).

| Model | PromptTTS2 | NaturalSpeech3 | VALL-E | VoiceBox | FlashSpeech | AudioGen | xTTS | Seed-TTS | OpenAI | Average |
|---|---|---|---|---|---|---|---|---|---|---|
| LFCC-LCNN | 0.6352 | 0.8604 | 0.8565 | 0.5699 | 0.8652 | 0.8743 | 0.7839 | 0.3889 | 0.6553 | 0.7210 |
| ResNet_Spec. | 0.7424 | 0.7090 | 0.6463 | 0.5734 | 0.7512 | 0.9646 | 0.5701 | 0.5734 | 0.6963 | 0.6920 |
| RawNet2 | 0.6368 | 0.5898 | 0.6757 | 0.3922 | 0.6994 | 0.9141 | 0.7351 | 0.4560 | 0.7082 | 0.6450 |
| RawGATST | 0.7458 | 0.6824 | 0.7793 | 0.4755 | 0.6011 | 0.9899 | 0.8043 | 0.5547 | 0.7073 | 0.7040 |
| AASIST | 0.7589 | 0.6797 | 0.7968 | 0.5218 | 0.6468 | 0.9344 | 0.7322 | 0.6706 | 0.6842 | 0.7140 |
| CLAP | 0.7216 | 0.9287 | 0.8250 | 0.3201 | 0.4497 | 0.8285 | 0.6924 | 0.3018 | 0.8247 | 0.6550 |
| Whisper-small | 0.9872 | 0.7803 | 0.7835 | 0.8159 | 0.4326 | 0.9159 | 0.8862 | 0.6771 | 0.2886 | 0.7300 |
| Whisper-large | 0.9924 | 0.8021 | 0.8568 | 0.8327 | 0.6283 | 0.9878 | 0.9218 | 0.6638 | 0.5107 | 0.8000 |
| Wave2Vec | 0.9664 | 0.7095 | 0.8351 | 0.9524 | 0.6439 | 0.9729 | 0.7113 | 0.9218 | 0.7290 | 0.8270 |
| HuBERT | 1.0000 | 0.8970 | 0.9155 | 0.9899 | 0.8477 | 1.0000 | 0.9197 | 0.9604 | 0.8169 | 0.9270 |
| Wave2VecBERT | 1.0000 | 0.6890 | 0.8632 | 0.9251 | 0.6239 | 0.9800 | 0.9665 | 0.5729 | 0.5289 | 0.7940 |

(c) EER(%) (↓).

| Model | PromptTTS2 | NaturalSpeech3 | VALL-E | VoiceBox | FlashSpeech | AudioGen | xTTS | Seed-TTS | OpenAI | Average |
|---|---|---|---|---|---|---|---|---|---|---|
| LFCC-LCNN | 44.0000 | 25.0000 | 21.0526 | 41.3462 | 23.7288 | 17.0000 | 28.3333 | 58.3333 | 37.6667 | 32.9400 |
| ResNet_Spec. | 32.0000 | 31.2500 | 38.9470 | 42.3080 | 31.3560 | 9.0000 | 43.8330 | 43.3330 | 35.0000 | 34.1140 |
| RawNet2 | 40.0000 | 43.7500 | 35.7895 | 56.7308 | 39.8305 | 13.0000 | 33.1667 | 54.1667 | 33.6667 | 38.9000 |
| RawGATST | 28.0000 | 34.7810 | 26.7633 | 55.4284 | 41.7860 | 2.0000 | 22.1833 | 47.8966 | 32.5000 | 32.3710 |
| AASIST | 27.0000 | 35.8720 | 23.3330 | 49.7712 | 38.6422 | 11.0000 | 31.4890 | 37.1550 | 36.7880 | 32.3390 |
| CLAP | 36.0000 | 18.7500 | 24.2105 | 66.3462 | 53.3898 | 24.0000 | 36.8333 | 66.0000 | 24.3333 | 38.8740 |
| Whisper-small | 4.0000 | 25.0000 | 25.2632 | 25.9615 | 55.9322 | 10.0000 | 19.5000 | 37.0000 | 64.6670 | 29.7030 |
| Whisper-large | 2.0000 | 23.0000 | 21.0732 | 21.7865 | 41.1744 | 4.0000 | 15.8943 | 37.8445 | 53.1270 | 24.4330 |
| Wave2Vec | 8.0000 | 28.1250 | 23.1579 | 9.6154 | 39.8305 | 6.0000 | 34.5000 | 12.6667 | 33.5000 | 21.7110 |
| HuBERT | 0.0000 | 18.7500 | 11.5789 | 3.8462 | 20.3390 | 0.0000 | 15.8333 | 9.3333 | 28.5000 | 12.0200 |
| Wave2VecBERT | 0.0000 | 37.5000 | 22.1053 | 11.5385 | 39.8305 | 2.0000 | 5.1667 | 44.5000 | 50.8333 | 23.7190 |

