# OpenReview forum: "SONAR: A Synthetic AI-Audio Detection Framework and Benchmark"
_ICLR.cc/2025/Conference — ICLR 2025 Conference Withdrawn Submission_

### Official Review · Reviewer_Rvoz · 2024-10-21

**Soundness:** 3
**Presentation:** 2
**Contribution:** 2
**Rating:** 3
**Confidence:** 5

**Summary:**

In this paper, the authors present SONAR, a new benchmark for AI-audio detection, derived from 9 TTS platforms. They claim this is the first framework to uniformly benchmark AI-audio detection across both traditional and foundation model-based deepfake detection systems. The authors also conduct extensive experiments to assess the model’s generalization capabilities and evaluate how factors such as model size and data scale affect model's performance.

**Strengths:**

1. The field of audio deepfake detection is an emerging area that has garnered significant attention.
2. The paper presents the ideas in a generally understandable way, with clear explanations of key concepts.
3. The authors are open about the study’s potential limitations and provide suggestions for future research directions.

**Weaknesses:**

1. **Related Works**: The paper does not adequately reference similar prior work in audio deepfake detection datasets. For instance, Voicewukong [1] offers a benchmark with more diverse data sourced from 34 speech synthesis platforms and includes multiple models, both traditional and advanced. Additionally, there are other works [2, 3] that also evaluate deepfake detectors using data from state-of-the-art generation methods, such as VALL-E and NaturalSpeech 3. Moreover, widely-used datasets like ASVspoof 2019 and 2021 [4, 5], although based on more traditional generation methods, are well-known benchmarks in this domain. It would strengthen the paper to acknowledge these datasets and clarify the specific advantages of SONAR compared to these established benchmarks.
2. **Dataset Construction**: The dataset presented (2274 fake samples) is relatively small compared to existing datasets, which typically contain more than 10,000 samples. Furthermore, the dataset is limited to 9 generation methods, with 6 of them sourced from demo pages or provided test set. The range of languages and voices is also quite restricted, which may limit the generalizability of the findings.
3. **Experiments**: While the authors conduct experiments to study generalization, many of the conclusions drawn are already well-established in previous research. For example, it is widely recognized that using foundation models significantly improves performance, and that increasing model size or adding data for fine-tuning enhances results. Additionally, due to the small size of the SONAR dataset, the results on each subset may not be sufficient to fully reflect the broader performance implications.

**References**:

[1] Yan, Z., Zhao, Y., & Wang, H. (2024). VoiceWukong: Benchmarking Deepfake Voice Detection. *arXiv preprint arXiv:2409.06348*.

[2] Xie, Y., Xiong, C., Wang, X., Wang, Z., Lu, Y., Qi, X., ... & Ye, L. (2024). Does Current Deepfake Audio Detection Model Effectively Detect ALM-based Deepfake Audio?. *arXiv preprint arXiv:2408.10853*.

[3] Li, Y., Zhang, M., Ren, M., Ma, M., Wei, D., & Yang, H. (2024). Cross-Domain Audio Deepfake Detection: Dataset and Analysis. *arXiv preprint arXiv:2404.04904*.

[4]  Nautsch, A., Wang, X., Evans, N., Kinnunen, T. H., Vestman, V., Todisco, M., ... & Lee, K. A. (2021). ASVspoof 2019: spoofing countermeasures for the detection of synthesized, converted and replayed speech. *IEEE Transactions on Biometrics, Behavior, and Identity Science*, *3*(2), 252-265.

[5] Liu, X., Wang, X., Sahidullah, M., Patino, J., Delgado, H., Kinnunen, T., ... & Lee, K. A. (2023). Asvspoof 2021: Towards spoofed and deepfake speech detection in the wild. *IEEE/ACM Transactions on Audio, Speech, and Language Processing*, *31*, 2507-2522.

**Questions:**

1. **Data Preprocessing**: In Figure 1, the data preprocessing is shown at the evaluation stage, but it’s unclear what preprocessing steps were applied to the data that was collected or generated. Since the data may vary in aspects such as sample rate and audio format, it would be helpful to explain the specific preprocessing methods used to ensure consistency across the dataset.
2. **AudioGen Data**: It is unclear why data from AudioGen, which consists of synthetic acoustic scenes, was included in SONAR, as these scenes don’t seem directly related to the paper’s proposed motivation. If the goal is to explore differences between detecting fake speech and synthetic sounds, it would be more effective to create two distinct subsets for speech and non-speech audio and incorporate a wider range of sound generation methods for a more thorough analysis.
3. **Training and Testing Settings**: The most common setting in deepfake detection involves training on the ASVspoof 2019 LA dataset [4] and testing on other datasets. It’s not clear why this setting wasn’t followed in the paper, or why test results on the ASVspoof dataset weren’t provided.

---

### Official Review · Reviewer_GGTJ · 2024-10-31

**Soundness:** 3
**Presentation:** 3
**Contribution:** 3
**Rating:** 6
**Confidence:** 4

**Summary:**

To assess the performance of speech forgery detection models against advanced TTS technologies, this paper proposes the SONAR framework. SONAR constructs a new evaluation dataset from nine different audio synthesis platforms based on the current leading TTS methods. Extensive experiments on this dataset are conducted using three metrics: EER, AUROC, and Accuracy, and subsequently analyzes the limitations of current detection methods in terms of generality. The aim is to provide an evaluation benchmark to promote forgery speech detection methods to cope with new forgery technologies. Such work is meaningful for current speech detection methods.

**Strengths:**

This paper introduces existing forgery speech detection methods, tests their performance against leading TTS-generated forgery speech, and the experiments are comprehensive. The proposed methods and the new evaluation dataset are a supplement to existing speech forgery detection research.

**Weaknesses:**

The dataset proposed in this work seems to focus only on TTS as a forgery method, but does not pay attention to forgery methods such as voice conversion and voice splicing, which is a deficiency of this work.

**Questions:**

see in weaknesses

---

### Official Review · Reviewer_aahH · 2024-11-03

**Soundness:** 3
**Presentation:** 3
**Contribution:** 2
**Rating:** 5
**Confidence:** 4

**Summary:**

The paper introduces an AI audio detection framework and benchmark named SONAR, designed to evaluate the generalization capability of existing AI-generated audio detection methods. A new evaluation dataset was constructed, comprising AI-generated audio samples from nine different audio synthesis platforms. Extensive experiments on both traditional and foundation models demonstrate the effectiveness of various models in detecting high-quality AI-generated audio. The results indicate that foundation models exhibit strong generalization capabilities, especially when pre-trained on a large scale. Additionally, the paper explores the effectiveness of few-shot fine-tuning in enhancing model generalization performance.

**Strengths:**

1. This study provides the  systematic evaluation of the generalization capabilities of various AI audio detection methods, introducing a rich and diverse dataset that helps bridge the gap in current research regarding generalization testing.

2. The framework includes multiple traditional and foundation models, and uses multidimensional evaluation metrics—such as accuracy, AUROC, and EER—to analyze the strengths and weaknesses of each model, offering a comprehensive benchmark.

3. The paper demonstrates the potential of few-shot fine-tuning to improve detection performance, making it especially suitable for customized or personalized detection systems.

4. By incorporating data from multiple audio synthesis platforms, the study covers a wide range of audio synthesis techniques, enhancing the framework's robustness for real-world applications.

**Weaknesses:**

This paper presents a substantial amount of work, collecting various state-of-the-art TTS-generated speech samples and conducting a comprehensive evaluation and comparative analysis of existing detection methods. The primary contributions lie in the construction of a novel dataset and the systematic testing of current approaches. However, from an innovation perspective, the paper appears relatively limited, as there are already several reviews and comparative studies on forgery speech detection methods. Therefore, it is suggested to further enhance the innovation of this paper on the basis of dataset construction and method testing.

1. Müller, Nicolas M., et al. "Mlaad: The multi-language audio anti-spoofing dataset." arXiv preprint arXiv:2401.09512 (2024).
2. Alali, Abdulazeez, and George Theodorakopoulos. "Review of existing methods for generating and detecting fake and partially fake audio." Proceedings of the 10th ACM International Workshop on Security and Privacy Analytics. 2024.
3. Chen, Tianxiang, et al. "Generalization of Audio Deepfake Detection." Odyssey. 2020.

**Questions:**

1. In the dataset constructed by the authors, there is significant variability in the number of samples across different types, resulting in an unbalanced distribution among categories (e.g., only 25 samples for PromptTTS2 compared to 600 for OpenAI). It would be helpful to understand the reasoning behind this design choice. Additionally, the dataset lacks details about the speakers and speech content, and the total sample size appears limited. These factors may significantly affect detection performance. Have the authors considered variables like speech encoding compression, format, and background noise? These aspects could impact the performance of forgery detection tasks, and increasing the dataset’s diversity and representativeness would improve the comparison of different models' robustness and generalization capabilities.

2. The MLAAD dataset includes 59 state-of-the-art TTS models, such as XTTS and Wishperspeech, covering multiple languages. Supplementing experiments with different models on this dataset would further enhance the credibility of the experimental conclusions.

Müller, Nicolas M., et al. "Mlaad: The multi-language audio anti-spoofing dataset." arXiv preprint arXiv:2401.09512 (2024).

3. In the setup of comparative experiments, aside from model architecture, did the authors consider the impact of techniques such as training data augmentation strategies and loss functions on model performance? These factors could play a significant role in shaping detection accuracy and robustness.

---

### Official Review · Reviewer_xc17 · 2024-11-26

**Soundness:** 3
**Presentation:** 3
**Contribution:** 1
**Rating:** 3
**Confidence:** 4

**Summary:**

The authors introduce a comprehensive evaluation framework for detecting AI-generated audio content, addressing the growing challenges posed by increasingly sophisticated text-to-speech (TTS) and voice conversion technologies. The authors also propose a dataset of 2274 samples using 9 latest speech generation methods.

**Strengths:**

- The paper introduces the first large-scale evaluation dataset that incorporates audio samples from nine different sources, including cutting-edge TTS providers and state-of-the-art models.
- Authors have done multiple experiments to evaluate the generalization capabilities of audio deepfake models.

**Weaknesses:**

- There exist Several audio deepfake datasets such as ASVSpoof5 that contain more diverse variations such as noise, compression, codecs  and multilingualism.
- The paper lacks documentation about the quality of the generated spoofed examples. In addition, the current antispoofing experiments are based on pre-trained speech model training, which lacks insights about which specific attributes in the speech may cause the model to fail.
- Authors should evaluate the generalization capabilities of the models using adding Room Impulse response in the audio.
- There should be additional experiment on how length of audio affect the deepfake detection capabilities.
-  Since I believe this is just an analysis paper, the research community would benefit from adding some interpretability and explainable methods such as Lime and SHapley [1]
- Authors mention in line (125) "We select 6 speakers from the LibriTTS dataset (Zen et al., 2019) as the reference speech and also
generate 600 text prompts with ChatGPT for each speaker, resulting in 600 synthetic speech audios.". I guess the total count would be 600 * 6 = 3600 audio. Author have mentioned it as 600 audio.


[1] Explaining deep learning models for spoofing and deepfake detection with SHapley Additive exPlanation

**Questions:**

Please see the weakness listed above.

---

### Note · Authors · 2024-11-26

I have read and agree with the venue's withdrawal policy on behalf of myself and my co-authors.